# BECAME: BayEsian Continual Learning with Adaptive Model MErging

Mei Li [* 1]  Yuxiang Lu [* 1]  Qinyan Dai [1]  Suizhi Huang [1]  Yue Ding [1]  Hongtao Lu [1]

## Abstract

Continual Learning (CL) strives to learn incrementally across tasks while mitigating catastrophic forgetting. A key challenge in CL is balancing stability (retaining prior knowledge) and plasticity (learning new tasks). While representative gradient projection methods ensure stability, they often limit plasticity. Model merging techniques offer promising solutions, but prior methods typically rely on empirical assumptions and carefully selected hyperparameters. In this paper, we explore the potential of model merging to enhance the stability-plasticity trade-off, providing theoretical insights that underscore its benefits. Specifically, we reformulate the merging mechanism using Bayesian continual learning principles and derive a closed-form solution for the optimal merging coefficient based on the Laplace approximation that adapts to the diverse characteristics of tasks. To validate our approach, we introduce a two-stage framework named BECAME, which synergizes the expertise of gradient projection and adaptive merging. Extensive experiments show that our approach outperforms state-of-the-art CL methods and existing merging strategies. Code is available at https://github.com/limei0818/BECAME.

## 1. Introduction

Continual Learning (CL) aims to equip models to learn incrementally, similar to human learning (Wang et al., 2024; De Lange et al., 2021; Mundt et al., 2023; Masana et al., 2022; Van de Ven et al., 2022). In the typical CL setting, models can only access the data of the current task due to privacy or storage constraints, yet they are expected to perform well on both the current and previously learned tasks. Since neural network models are typically optimized via

---
[*]Equal contribution [1]Shanghai Jiao Tong University. Correspondence to: Yue Ding <dingyue@sjtu.edu.cn>.

*Proceedings of the 42nd International Conference on Machine Learning*, Vancouver, Canada. PMLR 267, 2025. Copyright 2025 by the author(s).

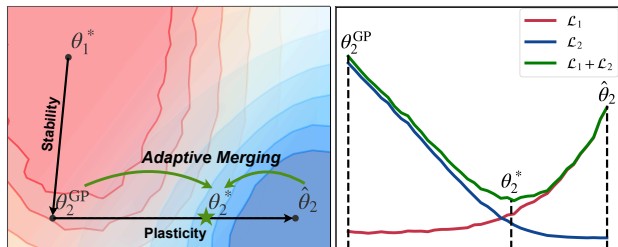

*Figure 1. (Left)* **The training loss landscape for task 1 and task 2**, represented by red and blue contours, respectively. Darker color denotes lower loss value. *(Right)* **The training loss values for task 1, task 2, and their sum.** Both figures are based on the NSCL-based experiment on the 10-split CIFAR-100 dataset. The model initially learns task 1, reaching $\theta_1^*$, which minimizes the loss $\mathcal{L}_1$. When learning task 2, the model first obtains $\theta_2^{\text{GP}}$ using the gradient projection, with minor forgetting indicated by the increase in $\mathcal{L}_1$. This solution shows limited plasticity, as $\mathcal{L}_2$ remains high. The model then proceeds to train without constraints, reaching $\hat{\theta}_2$, the minimum of $\mathcal{L}_2$. By analyzing the trajectory from $\theta_2^{\text{GP}}$ to $\hat{\theta}_2$, our method can determine the optimal merging coefficient, achieving the minimal cumulative loss at $\theta_2^*$.

gradient descent using gradients computed solely from the current task data, the loss on prior tasks can increase drastically (Hadsell et al., 2020). This phenomenon, known as catastrophic forgetting (McCloskey & Cohen, 1989; French, 1999), poses a major challenge in CL. Generally, CL models target to mitigate catastrophic forgetting by retaining knowledge from prior tasks (*stability*) while also being able to assimilate new tasks (*plasticity*). A balance between stability and plasticity is necessitated to achieve strong overall performance across both old and new tasks (Chaudhry et al., 2018; Mermillod et al., 2013; Kim & Han, 2023).

Projecting gradients of new data onto the subspace orthogonal to the feature space of previous tasks is a prominent method to ensure stability (Zeng et al., 2019; Wang et al., 2021b; Saha et al., 2021). Nevertheless, this approach often restricts plasticity, as it imposes strict constraints on parameter optimization, reducing the range of feasible gradient directions (Zhao et al., 2023). A promising solution involves merging the model's parameters with those of a complementary model specialized in the current task (Lin et al., 2022a). This idea of merging models of old and new tasks has also been empirically shown to enhance overall performance

across diverse scenarios (Lee et al., 2017; Marouf et al., 2024). However, these methods often acquire interpretation by assuming that learning new tasks is minimally influenced by previous knowledge, and then approximate the posterior of each task with an independent Gaussian distribution (Lee et al., 2017). Considering the sequential nature of learning, this assumption may not align with the context of CL, where tasks are inherently interrelated. Thus, the fundamental question remains to be answered: *How can model merging effectively improve the stability-plasticity trade-off in continual learning?*

Moreover, determining the merging coefficients for models is a critical challenge. A straightforward technique is to average the parameters of all models, as commonly adopted in prior work (Lin et al., 2022a; Lee et al., 2017; Marouf et al., 2024). Nonetheless, because different tasks often possess diverse characteristics, such as varying loss landscapes, assigning equal importance to all models can be suboptimal. While this limitation can be addressed by introducing tunable hyperparameters to adjust the importance of old and new tasks (Lee et al., 2017; Marouf et al., 2024), a major drawback lies in the complexity of this process and the risk of failing to find optimal parameters due to limited grid search. This naturally raises another question: *Can we derive the optimal solution in close-form for adaptively merging models?*

For the first question, we provide theoretical insights demonstrating that merging models can indeed yield a better optimum in CL (Section 3.2). Specifically, we show that along the path connecting the parameters of the old and new tasks, there always exists a point where the cumulative loss across all tasks is lower than at either endpoint. Furthermore, we reformulate the merging paradigm from the perspective of Bayesian continual learning (Kirkpatrick et al., 2017; Ritter et al., 2018; Kao et al., 2021), bridging the gap between prior work and the complexities of task interdependence. Regarding the second question, we aim to identify an optimal merging point that maximizes overall performance. To this end, we perform MAP estimation based on the posterior of sequentially learned tasks. We demonstrate that the optimization objective is convex along the linear merging path, enabling us to derive a closed-form solution for the optimal merging coefficient upon the Laplace approximation (Section 3.3). This solution adaptively integrates the parameters by considering the distinct loss landscapes of the tasks and the impact of parameter shifts, thereby achieving a balance between stability and plasticity.

To validate our findings, we introduce a two-stage framework named **BECAME**. The first stage involves optimizing the model using gradient projection (denoted by $\theta_2^{\text{GP}}$ in Figure 1), ensuring stability with only a minor increase in loss from the previous optimum ($\theta_1^*$). The model is then trained

without constraints to further enhance plasticity ($\hat{\theta}_2$). By synergizing the complementary strengths of these two models, our approach achieves superior performance compared to using either model independently, as indicated by the lowest cumulative loss in Figure 1 right. Moreover, our method offers a flexible framework that supports a wide variety of existing CL approaches, providing a general solution for consistent performance improvements.

In summary, our contributions are as follows:

- Capitalizing on Bayesian continual learning, we theoretically prove the existence of a merged model that adaptively balances the stability-plasticity trade-off and derives a closed-form solution for the optimal coefficient based on the Laplace approximation.

- We propose a simple yet effective method that not only substantiates our theoretical insights but also enhances the performance ceiling of various gradient projection methods, offering a general approach for exploiting plasticity.

- We evaluate our method on four widely-used continual learning benchmarks, showing that it remarkably outperforms previous state of the art, particularly in metrics measuring plasticity.

## 2. Related Works

**Continual Learning.** Existing Continual Learning (CL) methods can be broadly categorized into five main types: replay-based (Chaudhry et al., 2019b; Rebuffi et al., 2017; Schwarz et al., 2018), regularization-based (Kirkpatrick et al., 2017; Li & Hoiem, 2017; Ritter et al., 2018; Shin et al., 2017), architecture-based (Mallya & Lazebnik, 2018; Serra et al., 2018; Yoon et al., 2018; Hung et al., 2019; Liang & Li, 2023), representation-based (Javed & White, 2019; Pham et al., 2021; Madaan et al., 2022; Mehta et al., 2023), and optimization-based approaches (Zeng et al., 2019; Wang et al., 2021b; Saha et al., 2021) (Wang et al., 2024). Gradient projection methods, a subset of optimization-based approaches, achieve strong performance in mitigating forgetting by constraining updates to the orthogonal subspace of input features, while this may limit adaptability. Subsequent works (Saha & Roy, 2023; Lin et al., 2022b; Yang et al., 2024) have primarily focused on enhancing model plasticity.

**Model Merging.** Research on model merging generally falls into two categories: merging models fine-tuned on a single task to enhance overall generalization (Cha et al., 2021; Wortsman et al., 2022; Jordan et al., 2023), and merging models trained on distinct tasks to produce a unified multi-task model capable of handling all tasks from the

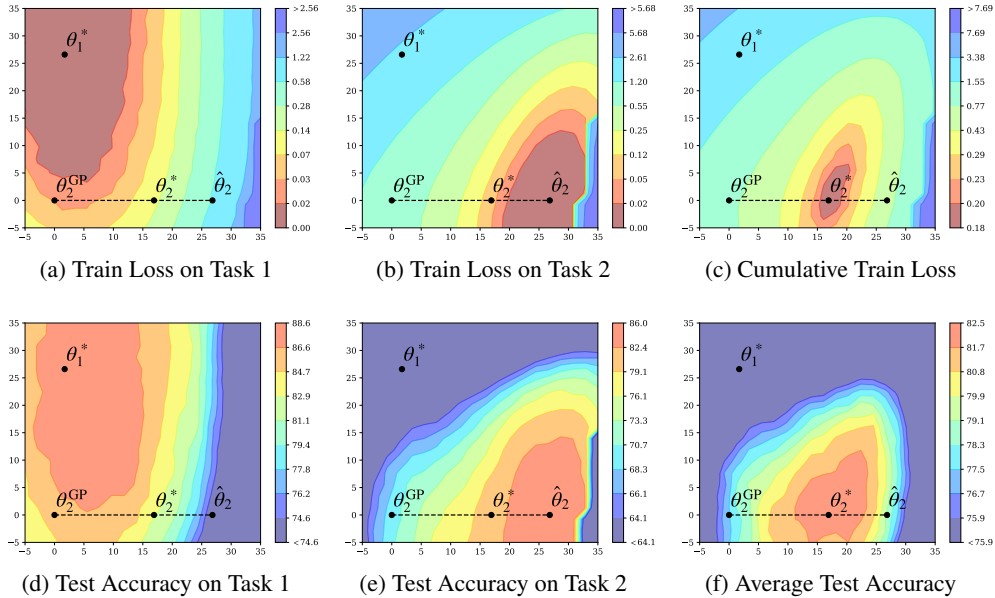

*Figure 2.* **Visualization of the training loss and test accuracy landscapes for task 1, task 2, cumulative training loss, and average test accuracy.** The figures are derived from the NSCL-based experiment conducted on the 10-split CIFAR-100 dataset. The model is initially trained on task 1, starting from a random initialization to obtain $\theta_1^*$. Next, the model undergoes training on task 2 with gradient projection, yielding $\theta_2^{\text{GP}}$. Further training on task 2 is then performed without constraints, resulting in $\hat{\theta}_2$. Our adaptive method then merges $\theta_2^{\text{GP}}$ and $\hat{\theta}_2$ to locate the optimal point that minimizes training loss along the trajectory from $\theta_2^{\text{GP}}$ to $\hat{\theta}_2$.

components (Matena & Raffel, 2022; Ilharco et al., 2023; Yadav et al., 2024). Model merging techniques are also applicable in continual learning (Stojanovski et al., 2022; Simon et al., 2022; Lee et al., 2020), where a critical challenge lies in determining the merging strategy. One intuitive approach involves using a constant weight related to the number of tasks (Lin et al., 2022a). Alternatively, IMM (Lee et al., 2017) and CoMA (Marouf et al., 2024) manually set coefficients through exhaustive tuning.

In contrast to existing merging methods for CL, we derive a closed-form solution that not only achieves optimal performance but also eliminates the need for manual adjustments across different scenarios. Additionally, our method is grounded in Bayesian continual learning, providing a robust theoretical foundation that aligns model merging with the core principles of CL.

## 3. Methodology

### 3.1. Preliminary

Under the setting of continual learning, a network $f_\theta$ with parameters $\theta \in \mathbb{R}^{|\theta|}$ is sequentially trained on $T$ tasks. Each task $t$ is defined by a subset of training samples $\mathcal{D}_t = \{(\mathbf{x}_t^{(n)}, \mathbf{y}_t^{(n)})\}_{n=1}^{|\mathcal{D}_t|}$ following a unique distribution $\mathbb{D}_t$, where $\mathbf{x}_t^{(n)}$ represents the input data and $\mathbf{y}_t^{(n)}$ the corresponding labels, with $t \in \{1, 2, \ldots, T\}$ denoting the task

identity. Each $\mathbb{D}_t$ is independent of others. The goal of continual learning is to achieve strong joint performance across all $T$ tasks after the model has been trained on each task sequentially. Specifically, while learning task $t$, only the current training data $\mathcal{D}_t$ is accessible, yet we aim to minimize the cumulative loss over all tasks learned so far $\mathcal{D}_{1:t} = \bigcup_{i=1}^{t} \mathcal{D}_i$:

$$\theta_t^* = \arg\min_\theta \mathcal{L}_{1:t}(\theta; \mathcal{D}_{1:t}) = \arg\min_\theta \sum_{i=1}^{t} \mathcal{L}_i(\theta; \mathcal{D}_i). \quad (1)$$

For simplicity, we denote $\mathcal{L}_i(\theta; \mathcal{D}_i)$ as $\mathcal{L}_i(\theta)$ and $\mathcal{L}_{1:t}(\theta; \mathcal{D}_{1:t})$ as $\mathcal{L}_{1:t}(\theta)$ in following sections.

Considering the nature of incremental learning, it is difficult to model the distribution of previous tasks $\{\mathbb{D}_1, \ldots, \mathbb{D}_{t-1}\}$ and new task $\mathbb{D}_t$ simultaneously when training the network on the new task, which leads to the critical challenge of stability-plasticity trade-off.

### 3.2. Merge Models for Better Minimum

To prove the effectiveness of model merging in continual learning, we begin with a standard setting. Suppose the model has been trained on tasks 1 to $t-1$, with its parameters $\theta_{t-1}^*$ representing the empirical optimal solution for the objective defined in Eq. 1. Next, the model is trained on task $t$ via a simple gradient descent optimizer, arriving at $\hat{\theta}_t$ by minimizing the task-specific loss $\mathcal{L}_t$. Given that the two

models are trained sequentially, their parameters are likely to reside in the same basin of the loss landscapes (Mirzadeh et al., 2021; Neyshabur et al., 2020). Generally, there should not be significant loss barriers along the linear path from $\theta_{t-1}^*$ to $\hat{\theta}_t$ when the parameters are weighted averaged. The interpolated model along this path could maintain high accuracy, comparable to both endpoints (Lee et al., 2017; Goodfellow et al., 2015; Marouf et al., 2024). In the following lemma, we prove that there exists a point on this trajectory where the merged model can achieve a better minimum for the cumulative loss overall learned tasks $\mathcal{L}_{1:t}$, indicating proficiency in both previous and new knowledge.

**Lemma 3.1.** *Given model parameters $\theta_{t-1}^*$, which minimize loss $\mathcal{L}_{1:t-1}$, and new parameters $\hat{\theta}_t$ optimized from $\theta_{t-1}^*$ to local minima of $\mathcal{L}_t$, there exists a coefficient $\lambda \in [0,1]$ satisfying that*

$$\mathcal{L}_{1:t}((1-\lambda)\theta_{t-1}^* + \lambda\hat{\theta}_t) \leq \min\{\mathcal{L}_{1:t}(\theta_{t-1}^*), \mathcal{L}_{1:t}(\hat{\theta}_t)\}. \quad (2)$$

*Proof.* With the merged parameters $\theta(\lambda) = (1-\lambda)\theta_{t-1}^* + \lambda\hat{\theta}_t$, we take the derivative of $\mathcal{L}_{1:t}(\theta(\lambda))$ with respect to $\lambda$:

$$\begin{aligned}
&\frac{\partial \mathcal{L}_{1:t}(\theta(\lambda))}{\partial \lambda} \\
&= \left(\frac{\partial \mathcal{L}_{1:t}(\theta(\lambda))}{\partial \theta(\lambda)}\right)^\top \cdot \frac{\partial \theta(\lambda)}{\partial \lambda} \quad (3) \\
&= \left(\frac{\partial \mathcal{L}_{1:t-1}(\theta(\lambda))}{\partial \theta(\lambda)} + \frac{\partial \mathcal{L}_t(\theta(\lambda))}{\partial \theta(\lambda)}\right)^\top (\hat{\theta}_t - \theta_{t-1}^*).
\end{aligned}$$

At $\lambda = 0$, $\theta(\lambda) = \theta_{t-1}^*$, given that $\frac{\partial \mathcal{L}_{1:t-1}}{\partial \theta}\big|_{\theta=\theta_{t-1}^*} = 0$ (as $\theta_{t-1}^*$ minimizes $\mathcal{L}_{1:t-1}$), we have

$$\frac{\partial \mathcal{L}_{1:t}(\theta(\lambda))}{\partial \lambda}\bigg|_{\lambda=0} = \left(\frac{\partial \mathcal{L}_t}{\partial \theta}\bigg|_{\theta=\theta_{t-1}^*}\right)^\top (\hat{\theta}_t - \theta_{t-1}^*) \leq 0, \quad (4)$$

because $\hat{\theta}_t$ is obtained by gradient descent from $\theta_{t-1}^*$, so the direction $\hat{\theta}_t - \theta_{t-1}^*$ is aligned with the negative gradient of $\mathcal{L}_t$ at the point $\theta_{t-1}^*$.

Similarly, when $\lambda = 1$, where $\theta(\lambda) = \hat{\theta}_t$, since the model converges to minima where gradient vanishes, *i.e.*, $\frac{\partial \mathcal{L}_t}{\partial \theta}\big|_{\theta=\hat{\theta}_t} = 0$, we have

$$\frac{\partial \mathcal{L}_{1:t}(\theta(\lambda))}{\partial \lambda}\bigg|_{\lambda=1} = \left(\frac{\partial \mathcal{L}_{1:t-1}}{\partial \theta}\bigg|_{\theta=\hat{\theta}_t}\right)^\top (\hat{\theta}_t - \theta_{t-1}^*) \geq 0, \quad (5)$$

as learning a new task generally increases the loss over previous tasks, making the direction from $\theta_{t-1}^*$ to $\hat{\theta}_t$ positively related to the gradient of $\mathcal{L}_{1:t-1}$ at $\hat{\theta}_t$.

Since $\mathcal{L}_{1:t}(\theta(\lambda))$ is continuous in $\lambda \in [0,1]$, with conditions on two endpoints of the merging path established in Eq. 4 and 5, there exists a coefficient $\lambda$ such that $\mathcal{L}_{1:t}(\theta(\lambda)) \leq \min\{\mathcal{L}_{1:t}(\theta_{t-1}^*), \mathcal{L}_{1:t}(\hat{\theta}_t)\}$. □

### 3.3. Determine Optimal Merging Coefficient

While Lemma 3.1 demonstrates that merging previous and current models can provide an improved solution for continual learning, determining the optimal merging coefficient remains a critical challenge. To address this, we derive a closed-form expression for the optimal coefficient upon the Laplace approximation by formulating the continual learning process from the perspective of Bayes' rule (Wang et al., 2024; Ritter et al., 2018; Kao et al., 2021).

The posterior after learning task $t$ is expressed as:

$$p(\theta|\mathcal{D}_{1:t}) \propto p(\theta)\prod_{i=1}^{t} p(\mathcal{D}_i|\theta) \propto p(\mathcal{D}_t|\theta)p(\theta|\mathcal{D}_{1:t-1}), \quad (6)$$

where $p(\theta)$ denotes the prior over the parameters. This equation implies that the posterior from previously observed tasks becomes the prior for task $t$, allowing the new posterior to be computed based solely on the new data.

According to the maximum a posteriori probability (MAP) estimation, the optimal parameters $\theta_t^*$ after learning task $t$ are

$$\begin{aligned}
\theta_t^* &= \arg\max_\theta \log p(\theta|\mathcal{D}_{1:t}) \\
&= \arg\max_\theta \log p(\mathcal{D}_t|\theta) + \log p(\theta|\mathcal{D}_{1:t-1}).
\end{aligned} \quad (7)$$

Alternatively, since the negative log likelihood $-\log p(\mathcal{D}_i|\theta)$ is typically used as the task-specific loss $\mathcal{L}_i(\theta)$, which can extend Eq. 7 as:

$$\begin{aligned}
\theta_t^* &= \arg\max_\theta \sum_{i=1}^{t} \log p(\mathcal{D}_i|\theta) + \log p(\theta) \\
&= \arg\min_\theta \sum_{i=1}^{t} \mathcal{L}_i(\theta),
\end{aligned} \quad (8)$$

where the log prior $\log p(\theta)$ is not related to the optimization when it follows a uniform distribution. Eq. 8 indicates that the objective formed by MAP is equivalent to minimizing the cumulative loss introduced in Eq. 1.

As the posterior $p(\theta|\mathcal{D}_{1:t-1})$ is generally intractable, it is a common strategy to use Laplace approximation (MacKay, 1992), approximating it by a multivariate Gaussian: $p(\theta|\mathcal{D}_{1:t-1}) \approx q_{t-1}(\theta) = \mathcal{N}(\theta; \mu_{t-1}, \Lambda_{t-1}^{-1})$ (Huszár, 2018; Kirkpatrick et al., 2017; Ritter et al., 2018), where $\Lambda_{t-1}$ is positive semi-definite. Similarly, $p(\theta|\mathcal{D}_{1:t}) \approx q_t(\theta) = \mathcal{N}(\theta; \mu_t, \Lambda_t^{-1})$. Thus, we can rewrite the MAP in Eq. 7 as:

$$\theta_t^* = \arg\max_\theta \log p(\mathcal{D}_t|\theta) - \frac{1}{2}(\theta - \mu_{t-1})^\top \Lambda_{t-1}(\theta - \mu_{t-1}), \quad (9)$$

where $\mu_{t-1} = \theta_{t-1}^*$ represents the MAP parameters after learning task $t-1$. Since the MAP is often optimized by minimizing the loss function, Eq. 9 can be reformulated as:

$$\theta_t^* = \arg \min_\theta \mathcal{L}_t(\theta) + \frac{1}{2}(\theta - \theta_{t-1}^*)^\top \Lambda_{t-1}(\theta - \theta_{t-1}^*). \quad (10)$$

Next, we substitute the merged parameters $\theta(\lambda) = (1 - \lambda)\theta_{t-1}^* + \lambda \hat{\theta}_t$ into Eq. 10. Defining $\Delta\theta = \hat{\theta}_t - \theta_{t-1}^*$, the objective becomes:

$$\tilde{\mathcal{L}}_{1:t}(\lambda) = \mathcal{L}_t\left(\hat{\theta}_t + (\lambda - 1)\Delta\theta\right) + \frac{1}{2}\lambda^2 \Delta\theta^\top \Lambda_{t-1}\Delta\theta. \quad (11)$$

Then we perform a second-order Taylor expansion of the first term around $\hat{\theta}_t$,

$$\begin{aligned}
\mathcal{L}_t(\theta(\lambda)) &\approx \mathcal{L}_t(\hat{\theta}_t) + (\lambda - 1)\Delta\theta^\top \nabla_\theta \mathcal{L}_t|_{\theta=\hat{\theta}_t} \\
&\quad + \frac{1}{2}(\lambda - 1)^2 \Delta\theta^\top \nabla_\theta^2 \mathcal{L}_t|_{\theta=\hat{\theta}_t}\Delta\theta \\
&\approx \mathcal{L}_t(\hat{\theta}_t) + \frac{1}{2}(\lambda - 1)^2 \Delta\theta^\top \nabla_\theta^2 \mathcal{L}_t|_{\theta=\hat{\theta}_t}\Delta\theta,
\end{aligned} \quad (12)$$

where $\nabla_\theta \mathcal{L}_t$ is the gradient and $\nabla_\theta^2 \mathcal{L}_t$ is the Hessian with respect to $\theta$. As $\hat{\theta}_t$ represents the parameters at local minima of $\mathcal{L}_t$, the gradient $\nabla_\theta \mathcal{L}_t|_{\theta=\hat{\theta}_t}$ equals zero, and the Hessian $\nabla_\theta^2 \mathcal{L}_t|_{\theta=\hat{\theta}_t}$ is positive semi-definite.

Taking the derivative with respect to $\lambda$, we have:

$$\frac{\partial \tilde{\mathcal{L}}_{1:t}(\lambda)}{\partial \lambda} = (\lambda - 1)\Delta\theta^\top \nabla_\theta^2 \mathcal{L}_t|_{\theta=\hat{\theta}_t}\Delta\theta + \lambda \Delta\theta^\top \Lambda_{t-1}\Delta\theta. \quad (13)$$

Considering the derivatives at the endpoints $\lambda = 0$ and $\lambda = 1$, we have:

$$\left.\frac{\partial \tilde{\mathcal{L}}_{1:t}(\lambda)}{\partial \lambda}\right|_{\lambda=0} = -\Delta\theta^\top \nabla_\theta^2 \mathcal{L}_t|_{\theta=\hat{\theta}_t}\Delta\theta \leq 0, \quad (14)$$

$$\left.\frac{\partial \tilde{\mathcal{L}}_{1:t}(\lambda)}{\partial \lambda}\right|_{\lambda=1} = \Delta\theta^\top \Lambda_{t-1}\Delta\theta \geq 0. \quad (15)$$

We also consider the second-order derivative:

$$\frac{\partial^2 \tilde{\mathcal{L}}_{1:t}(\lambda)}{\partial \lambda^2} = \Delta\theta^\top \nabla_\theta^2 \mathcal{L}_t|_{\theta=\hat{\theta}_t}\Delta\theta + \Delta\theta^\top \Lambda_{t-1}\Delta\theta \geq 0, \quad (16)$$

indicating that $\tilde{\mathcal{L}}_{1:t}(\lambda)$ is convex along the merging path. With the property that the derivative is opposite at two endpoints, there exists a unique optimal $\lambda^*$ in $[0,1]$ that minimizes the objective when the derivative in Eq. 13 equals zero, yielding the closed-form solution:

$$\lambda_t^* = \frac{\Delta\theta^\top \nabla_\theta^2 \mathcal{L}_t|_{\theta=\hat{\theta}_t}\Delta\theta}{\Delta\theta^\top (\nabla_\theta^2 \mathcal{L}_t|_{\theta=\hat{\theta}_t} + \Lambda_{t-1})\Delta\theta}. \quad (17)$$

Then we continue to introduce how to calculate $\lambda_t^*$ in practice. By performing a quadratic approximation of $p(\theta|\mathcal{D}_{1:t})$,

**Algorithm 1** Pseudo-codes for **BECAME**

**Input:** $T$ sequential tasks with data $\{\mathcal{D}_1, \mathcal{D}_2, \cdots \mathcal{D}_T\}$, model $f_\theta$ initialized with parameters $\theta_0$
**Output:** Trained model parameters $\theta_T^*$
1: Train on task $\mathcal{D}_1$ to get $\theta_1^*$
2: Compute precision matrix with Fisher Information $\Lambda_1 = F_1(\theta_1^*)$
3: **for** $t = 2, 3, \ldots, T$ **do**
4:     Train on task $\mathcal{D}_t$ with gradient projection to get $\theta_t^{\text{GP}}$
5:     Train on task $\mathcal{D}_t$ without constraint to get $\hat{\theta}_t$
6:     Compute the current Fisher Information $F_t(\hat{\theta}_t)$
7:     Derive merging coefficient $\lambda_t^*$ via Eq. 20
8:     Merge models $\theta_t^* = (1 - \lambda_t^*)\theta_t^{\text{GP}} + \lambda_t^* \hat{\theta}_t$
9:     Update precision matrix $\Lambda_t = F_t(\theta_t^*) + \Lambda_{t-1}$
10: **end for**

the precision matrix $\Lambda_t$ can be computed as the Hessian of the negative log posterior:

$$\begin{aligned}
\Lambda_t &= -\nabla_\theta^2 \log p(\theta|\mathcal{D}_{1:t})|_{\theta=\mu_t} \\
&\approx -\nabla_\theta^2 \log p(\mathcal{D}_t|\theta)|_{\theta=\mu_t} + \Lambda_{t-1},
\end{aligned} \quad (18)$$

where the first term is the Hessian of the negative log likelihood at $\mu_t = \theta_t^*$. Eq. 18 means that the precision matrix is updated recursively, initialized by the Hessian of the negative log prior, which is typically constant.

Nevertheless, computing the Hessian can be computationally expensive due to the large scale of model parameters. Moreover, for the precision $\Lambda_t$ to be valid, it is required to be positive semi-definite, which is not always guaranteed when approximating with the Hessian (Wang et al., 2024; Ritter et al., 2018). To address these limitations, the Fisher Information Matrix (FIM) is used as an approximation of the Hessian (Geisser et al., 1990; Martens, 2020; Kirkpatrick et al., 2017; Ritter et al., 2018), as it is positive semi-definite by construction:

$$F_t(\theta) = \mathbb{E}[\nabla_\theta \log p(\mathcal{D}_t|\theta)\nabla_\theta \log p(\mathcal{D}_t|\theta)^\top]. \quad (19)$$

Therfore, $\lambda_t^*$ can be calculated by the following equation:

$$\lambda_t^* = \frac{\Delta\theta^\top F_t(\hat{\theta}_t)\Delta\theta}{\Delta\theta^\top (F_t(\hat{\theta}_t) + \sum_{i=1}^{t-1} F_i(\theta_i^*))\Delta\theta}. \quad (20)$$

Furthermore, the FIM can be simplified with a diagonal approximation (Huszár, 2018; Kirkpatrick et al., 2017).

### 3.4. BECAME

In Lemma 3.1, we establish an upper bound for the cumulative loss after merging, $\mathcal{L}_{1:t}(\theta(\lambda_t^*))$, at the lower value of the two endpoints on the merging path. To further tighten this bound, we introduce a two-stage training paradigm. In

*Table 1.* **Performance comparison of GPM-based experiments.** * denote the results from previous work. We reproduce GPM, TRGP, SGP, and GPCNS under our setting for a fair comparison. The best ACC is marked in **bold**, and the second-best is underlined.

| Method | 20-Split CIFAR-100 | | 10-Split CIFAR-100 | | 20-Split MiniImageNet | |
|---|---|---|---|---|---|---|
| | ACC(%)↑ | BWT(%)↑ | ACC(%)↑ | BWT(%)↑ | ACC(%)↑ | BWT(%)↑ |
| Multi-task | 85.24 ± 0.59 | - | 79.52 ± 0.59 | - | 77.78 ± 1.49 | - |
| EWC* (Kirkpatrick et al., 2017) | 75.30 ± 0.70 | -6.30 ± 0.60 | 68.80 ± 0.88 | -20.00 ± 1.00 | 52.10 ± 1.10 | -9.30 ± 1.40 |
| A-GEM* (Chaudhry et al., 2019a) | - | - | 63.98 ± 1.22 | -15.00 ± 2.00 | 57.24 ± 0.72 | -12.00 ± 1.00 |
| FS-DGPM* (Wang et al., 2021a) | 80.50 ± 0.40 | -3.30 ± 0.40 | 74.33 ± 0.31 | -3.00 ± 0.00 | | |
| ER-Res* (Chaudhry et al., 2019b) | 79.20 ± 0.40 | -4.90 ± 0.50 | 71.73 ± 0.63 | -6.00 ± 1.00 | 55.20 ± 2.90 | -5.70 ± 0.80 |
| GPM (Saha et al., 2021) | 77.34 ± 0.55 | 0.01 ± 0.07 | 71.81 ± 0.10 | -0.11 ± 0.33 | 63.90 ± 1.05 | -1.30 ± 1.27 |
| GPM + Ours | 80.57 ± 0.18 | -0.06 ± 0.15 | 75.05 ± 0.27 | 0.02 ± 0.39 | 67.62 ± 1.40 | 0.87 ± 0.26 |
| TRGP (Lin et al., 2022b) | 81.68 ± 0.69 | -0.13 ± 0.15 | 75.01 ± 0.63 | -0.01 ± 0.16 | 62.68 ± 2.49 | -1.04 ± 0.78 |
| TRGP + Ours | **82.61** ± 0.38 | -0.32 ± 0.18 | 75.87 ± 0.43 | -0.77 ± 0.27 | 65.09 ± 1.81 | -0.65 ± 1.08 |
| SGP (Saha & Roy, 2023) | 80.21 ± 0.68 | -0.88 ± 0.29 | 74.97 ± 0.31 | -0.98 ± 0.30 | 66.99 ± 2.33 | -2.46 ± 1.28 |
| SGP + Ours | 81.94 ± 0.40 | -1.22 ± 0.25 | **76.74** ± 0.36 | -1.42 ± 0.25 | **70.06** ± 2.12 | -0.26 ± 0.72 |
| GPCNS (Yang et al., 2024) | 78.63 ± 0.63 | -1.93 ± 0.42 | 71.84 ± 0.93 | -3.44 ± 0.85 | 62.85 ± 2.18 | -1.90 ± 1.75 |
| GPCNS + Ours | 80.87 ± 0.24 | -1.96 ± 0.24 | 73.89 ± 0.52 | -3.46 ± 0.68 | 64.79 ± 1.71 | -2.07 ± 1.66 |

the first stage, the model is initialized from the last state $\theta_{t-1}^*$, and trained with gradient projection to obtain $\theta_t^{\text{GP}}$. Then in the second stage, we continue training from $\theta_t^{\text{GP}}$ without constraints to reach $\hat{\theta}_t$.

The key idea of gradient projection is to maintain the network's outputs when fed inputs from previously seen tasks (Wang et al., 2021b; Saha et al., 2021; Saha & Roy, 2023), leading to nearly identical output logits and minimal influence on the losses of previous tasks, *i.e.*, $\mathcal{L}_{1:t-1}(\theta_t^{\text{GP}}) \approx \mathcal{L}_{1:t-1}(\theta_{t-1}^*)$. Thus, we can assume that the projected model remains at the local minima of the previous losses:

$$\frac{\partial \mathcal{L}_{1:t-1}}{\partial \theta}\bigg|_{\theta=\theta_t^{\text{GP}}} \approx \frac{\partial \mathcal{L}_{1:t-1}}{\partial \theta}\bigg|_{\theta=\theta_{t-1}^*} = 0, \qquad (21)$$

which implies that Lemma 3.1 stills holds when replacing the starting point $\theta_{t-1}^*$ on the path with $\theta_t^{\text{GP}}$. Additionally, we can approximate the posterior of previous tasks using the new parameters, facilitating a smooth loss surface around the path (Lee et al., 2017; Marouf et al., 2024). This allows us to substitute $\Delta\theta$ with $\hat{\theta}_t - \theta_t^{\text{GP}}$ when determining the optimal coefficient in Eq. 20.

*Proof.* To justify this substitution theoretically, we prove that Eq. 10 holds when $\theta_{t-1}^*$ is replaced by $\theta_t^{\text{GP}}$.

Given that $\Lambda_{t-1}$ is symmetric, we can expand the second term $(\theta - \theta_t^{\text{GP}})^\top \Lambda_{t-1}(\theta - \theta_t^{\text{GP}})$ in Eq. 10 as:

$$(\theta - \theta_{t-1}^*)^\top \Lambda_{t-1}(\theta - \theta_{t-1}^*)$$
$$+ \big(2(\theta - \theta_{t-1}^*) + (\theta_{t-1}^* - \theta_t^{\text{GP}})\big)^\top \Lambda_{t-1}(\theta_{t-1}^* - \theta_t^{\text{GP}}). \quad (22)$$

Then we demonstrate that $\Lambda_{t-1}(\theta_{t-1}^* - \theta_t^{\text{GP}}) = 0$, which can result in that the second term in Eq. 22 is zero so that $\theta_{t-1}^*$ can be replaced by $\theta_t^{\text{GP}}$ in Eq. 10.

Perform a second-order Taylor expansion of $\mathcal{L}_{1:t-1}(\theta_t^{\text{GP}})$ around $\theta_{t-1}^*$ and let $d = \theta_t^{\text{GP}} - \theta_{t-1}^*$, there is:

$$\mathcal{L}_{1:t-1}(\theta_t^{\text{GP}}) \approx \mathcal{L}_{1:t-1}(\theta_{t-1}^*) + d^\top \nabla_\theta \mathcal{L}_{1:t-1}|_{\theta=\theta_{t-1}^*}$$
$$+ \frac{1}{2}d^\top \nabla_\theta^2 \mathcal{L}_{1:t-1}|_{\theta=\theta_{t-1}^*}d, \quad (23)$$

where $\nabla_\theta \mathcal{L}_{1:t-1}|_{\theta=\theta_{t-1}^*} = 0$, since $\theta_{t-1}^*$ is an optimum for $\mathcal{L}_{1:t-1}$. Moreover, as there is $\mathcal{L}_{1:t-1}(\theta_t^{\text{GP}}) \approx \mathcal{L}_{1:t-1}(\theta_{t-1}^*)$, we have

$$d^\top \nabla_\theta^2 \mathcal{L}_{1:t-1}|_{\theta=\theta_{t-1}^*}d = 0. \quad (24)$$

Recall that $\mathcal{L}_{1:t-1} = -\log p(\theta|\mathcal{D}_{1:t-1})$ (Eq. 8) and $\Lambda_{t-1} = -\nabla_\theta^2 \log p(\theta|\mathcal{D}_{1:t-1})|_{\theta=\theta_{t-1}^*}$ (Eq. 18), we can replace $\nabla_\theta^2 \mathcal{L}_{1:t-1}|_{\theta=\theta_{t-1}^*}$ with $\Lambda_{t-1}$ to get

$$d^\top \Lambda_{t-1}d = 0. \quad (25)$$

Moreover, by decomposing $\Lambda_{t-1}$ via eigendecomposition as $\Lambda_{t-1} = Q^\top A Q$, where $A$ is a diagonal matrix containing the eigenvalues, there is $d^\top \Lambda_{t-1}d = d^\top Q^\top A Q d = 0$.

Since $\Lambda_{t-1}$ is positive semi-definite, the eigenvalues $A_{ii} \geq 0$ require that each element of $Qd$ must be zero, leading to $\Lambda_{t-1}d = (Q^\top A)(Qd) = 0$. Therefore, we can get $\Lambda_{t-1}(\theta_t^{\text{GP}} - \theta_{t-1}^*) = 0$, and the second term in Eq. 22 is zero. Thus, we have:

$$(\theta - \theta_t^{\text{GP}})^\top \Lambda_{t-1}(\theta - \theta_t^{\text{GP}}) = (\theta - \theta_{t-1}^*)^\top \Lambda_{t-1}(\theta - \theta_{t-1}^*) \quad (26)$$

This equality validates our substitution of $\Delta\theta$ with $\hat{\theta}_t - \theta_t^{\text{GP}}$ in Eq. 20. □

Additionally, as the loss of new task is substantially reduced compared to $\theta_{t-1}^*$ through training, it can entail a lower

*Table 2.* **Performance comparison of NSCL-based experiments.** * denotes results from Connector (Lin et al., 2022a), which does not report the standard deviation. We reproduce Adam-NSCL under our setting for a fair comparison.

| Method | 20-Split CIFAR-100 | | 10-Split CIFAR-100 | | 25-Split TinyImageNet | |
|---|---|---|---|---|---|---|
| | ACC(%)↑ | BWT(%)↑ | ACC(%)↑ | BWT(%)↑ | ACC(%)↑ | BWT(%)↑ |
| Multi-task | $90.81 \pm 0.10$ | - | $95.06 \pm 0.18$ | - | $82.55 \pm 0.12$ | - |
| OWM* (Zeng et al., 2019) | 68.47 | -3.37 | 68.89 | -1.88 | 49.98 | -3.64 |
| EWC* (Kirkpatrick et al., 2017) | 71.66 | -3.72 | 70.77 | -2.83 | 52.33 | -6.17 |
| SI* (Ostapenko et al., 2019) | 59.76 | -8.62 | 60.57 | -5.17 | 45.27 | -4.45 |
| LwF* (Li & Hoiem, 2017) | 74.38 | -9.11 | 70.70 | -6.27 | 56.57 | -11.19 |
| GD-WILD* (Lee et al., 2019) | 77.16 | -14.85 | 71.27 | -18.24 | 42.74 | -34.58 |
| Connector* (Lin et al., 2022a) | 80.80 | -5.00 | 79.79 | -0.92 | 64.61 | -6.00 |
| Adam-NSCL (Wang et al., 2021b) | $75.66 \pm 0.39$ | $-3.85 \pm 0.45$ | $72.91 \pm 0.30$ | $-1.76 \pm 0.20$ | $58.57 \pm 0.40$ | $-5.74 \pm 0.61$ |
| Adam-NSCL + Ours | $\mathbf{81.88} \pm 0.37$ | $-4.56 \pm 0.42$ | $\mathbf{81.66} \pm 0.51$ | $-0.80 \pm 0.53$ | $\mathbf{66.49} \pm 0.48$ | $-4.97 \pm 0.73$ |

cumulative loss $\mathcal{L}_{1:t}(\theta_t^{\text{GP}}) < \mathcal{L}_{1:t}(\theta_{t-1}^*)$. This guarantees a better upper bound on the merged model's loss, which is consistently lower than that of the projected model, resulting in an improved minimum after merging with the second-stage model. The complete pipeline of **BECAME** is outlined in Algorithm 1.

## 4. Experiments

### 4.1. Experimental Setup

**Baselines.** Our experiments are divided into two categories. The first category is based on GPM (Saha et al., 2021) and its extensions: TRGP (Lin et al., 2022b), SGP (Saha & Roy, 2023), and GPCNS (Yang et al., 2024). We refer to these as "GPM-based experiments". The second category is based on Adam-NSCL (NSCL) (Wang et al., 2021b), which we similarly term "NSCL-based experiments". In addition to these baseline methods, we also compare our approach with OWM (Zeng et al., 2019), Connector (Lin et al., 2022a), EWC (Kirkpatrick et al., 2017), SI (Ostapenko et al., 2019), LwF (Li & Hoiem, 2017), GD-WILD (Lee et al., 2019), A-GEM (Chaudhry et al., 2019a), FS-DGPM (Wang et al., 2021a), and ER-Res (Chaudhry et al., 2019b). Appendix B.1 introduces these methods in more detail.

**Datasets.** We conduct our experiments on four widely-used benchmarks: 20-Split CIFAR-100 (Krizhevsky et al., 2009), 10-Split CIFAR-100, 25-Split TinyImageNet (Wu et al., 2017), and 20-Split MiniImageNet (Vinyals et al., 2016). *k-Split* means the dataset is evenly divided into $k$ tasks. Details can be found in Appendix B.2.

**Architecture.** To ensure a fair comparison, we use the same network architectures as those employed in the baselines. For GPM-based experiments, we utilize a 5-layer AlexNet-like network as the backbone for the 10-Split and 20-Split CIFAR-100 datasets, and a reduced version of ResNet-18 (He et al., 2016) for the 20-Split MiniImageNet experiment. For NSCL-based experiments, we employ ResNet-18 across all datasets. Each task has a separate classifier head under the task-incremental learning setup.

**Implementation Details.** For the first task, the training process is identical to that of the corresponding baselines. For subsequent tasks $t \in \{2, 3, \cdots, T\}$, our method involves two stages as mentioned above. All experiments are repeated with 5 random seeds, and we report the mean and standard deviation of the results. The hyperparameter configurations in both stages are mostly consistent with those of the baselines except subtle adjustments for adapting our methods, with details provided in Appendix B.4 to ensure reproducibility.

**Metrics.** We use two evaluation metrics for the main experimental results: average accuracy (ACC) and backward transfer (BWT) (Lopez-Paz & Ranzato, 2017). ACC measures the overall test performance across all learned tasks, while BWT measures the stability. Besides, we use the Intransigence Measure (IM) (Chaudhry et al., 2018) to assess plasticity. Definition of the metrics can be found in Appendix B.5.

### 4.2. Main Results

**Visualization.** In Figure 2, we present a detailed visualization of the training loss and test accuracy landscapes for the first two tasks of the 10-split CIFAR-100 dataset. The plots display the performance of different model parameter configurations, specifically $\theta_1^*$, $\theta_2^{\text{GP}}$, $\hat{\theta}_2$, and $\theta_2^*$. The gap between $\theta_2^{\text{GP}}$ and $\hat{\theta}_2$ illustrates the limited plasticity of gradient projection, with the path from $\theta_2^{\text{GP}}$ to $\hat{\theta}_2$ representing the stability-plasticity trade-off. The plots clearly demonstrate that the merged model $\theta_2^*$ achieves the optimal balance between the performance on both tasks, as shown in panels (c) and (f). The placement of $\theta_2^*$ along the path suggests that it precisely identifies the optimal merging point, minimizing the optimization objective and significantly outperforming both $\theta_2^{\text{GP}}$ and $\hat{\theta}_2$ in overall performance.

*Table 3.* **Performance comparison of different merging strategies on 10-Split CIFAR-100 (10-S CIFAR) and 20-Split MiniImageNet (20-S Mini) based on GPM.** The results demonstrate that model merging enhances overall performance. Our method strikes a superior balance by maximizing plasticity while minimizing forgetting.

| Dataset | Strategy | ACC(%)↑ | BWT(%)↑ |
|---|---|---|---|
| | - | 71.81 | -0.11 |
| 10-S | $1/t$ (Lee et al., 2017) | 74.71 | -0.40 |
| CIFAR | CoMA (Marouf et al., 2024) | 74.07 | -2.79 |
| | CoFiMA (Marouf et al., 2024) | 74.36 | -1.77 |
| | Ours | **75.05** | **0.02** |
| | - | 63.90 | -1.30 |
| 20-S | $1/t$ (Lee et al., 2017) | 66.09 | 0.57 |
| Mini | CoMA (Marouf et al., 2024) | 66.91 | -0.82 |
| | CoFiMA (Marouf et al., 2024) | 64.19 | 0.54 |
| | Ours | **67.62** | **0.87** |

**Comparison with state of the art.** Table 1 presents the results of the GPM-based experiments. Our method consistently achieves higher accuracy across all three datasets compared to four baselines. Notably, it boosts the accuracy of GPM by over 3% on all datasets. Beyond improving the performance of the corresponding baseline, our method applied to GPM also surpasses TRGP, SGP, and GPCNS, which are alternative GPM improvement methods, on the 10-Split CIFAR-100 and 20-Split MiniImageNet datasets. Furthermore, the increase in accuracy is achieved without compromising the ability to mitigate forgetting, demonstrating a well-maintained stability-plasticity trade-off.

Table 2 reports the results of the NSCL-based experiments. As shown in the table, our method remarkably enhances the performance of NSCL. On 20-Split CIFAR-100, our method improves ACC by over 6%. For two more challenging datasets, 10-Split CIFAR-100 and 25-Split TinyImageNet, the improvement is even more pronounced, with an ACC increase of 8%. Additionally, our approach outperforms Connector in all experiments, which is regarded as the state of the art among NSCL-based methods.

### 4.3. In-depth Analysis

**Merging Strategy.** To validate the effectiveness of our adaptive merging, we compare it with existing merging strategies proposed by Connector (Lin et al., 2022a), IMM (Lee et al., 2017), and CoMA (Marouf et al., 2024). These methods typically merge models using a fixed coefficient based on the number of learned tasks ($1/t$) or a tunable constant based on grid search. CoFiMA (Marouf et al., 2024) further introduces a parameter-wise Fisher-weighted averaging with a tunable hyperparameter.

Since Connector (Lin et al., 2022a) is also based on NSCL, we can directly compare it with our approach. As shown in Table 2, our method consistently surpasses Connector

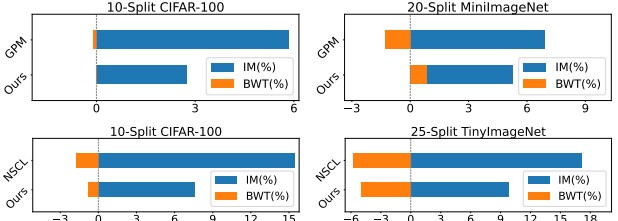

*Figure 3.* **BWT and IM in GPM-based and NSCL-based experiments.** Our method outperforms baselines by reducing IM (plasticity, lower is better) while maintaining a good BWT (stability, higher is better).

across all three benchmarks. IMM is reported on different benchmarks, and CoMA/CoFiMA rely on a pre-trained model, which is a distinct setting in CL. To ensure a fair and accurate comparison, we implement their merging strategies within our framework and use the hyperparameters reported in their papers. Table 3 shows that all merging strategies improve accuracy over the baseline without merging, providing empirical support for our theoretical insights. Among these strategies, our adaptive coefficient achieves the highest ACC and performs well in terms of BWT. In contrast, CoMA, which uses a fixed coefficient, falls short in BWT as it tends to prioritize the most recent tasks. Performance comparison without gradient projection training stage is available in Appendix C.3.

**Plasticity-Stability Trade-off.** For a specific task $i$, we can find the relationship between the final accuracy $A_{T,i}$, $BWT_i$, and $IM_i$, formulated as:

$$A_{T,i} = A_i^* - IM_i + BWT_i, \qquad (27)$$

where $BWT_i$ and $IM_i$ are calculated individually for each task. Since $A_i^*$ remains constant for a given backbone regardless of the chosen approach, $A_{T,i}$ depends on $IM_i$ and $BWT_i$. Optimal performance is achieved when IM is minimized (*i.e.*, better plasticity) and BWT is maximized (*i.e.*, better stability). Figure 3 illustrates this trade-off by visualizing BWT and IM using horizontal bars. The length of each bar represents the absolute value of the respective metric. By substantially minimizing IM, our method outperforms both GPM and NSCL, enhancing plasticity towards the upper bound while preserving stability across different scenarios.

**Balance Across Tasks.** In experimental studies, it is often assumed that all tasks hold equal importance during testing. Nonetheless, in real-world applications, task importance can vary due to difference in the number of samples per task. This variation highlights the necessity of achieving inter-task balance. To evaluate this, we include the standard deviation of the final accuracy (STD) as an additional metric. Considering both ACC and STD is essential: when methods attain similar ACC, a lower STD reflects less bias among

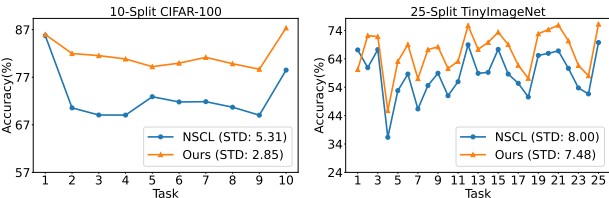

*Figure 4.* **Final test accuracy of each task in GPM-based and NSCL-based experiments.** Our method simultaneously enhances the accuracy and balance across different tasks (STD).

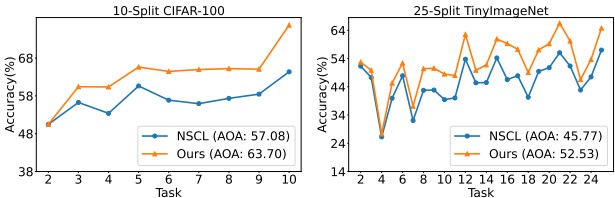

*Figure 5.* **After one epoch accuracy (AOA) in GPM-based and NSCL-based experiments.** Note that the training process is the same until the second stage of task 2. Our methods demonstrate superior generalization for learning new tasks compared to the baselines.

tasks. As shown in Figure 4, the accuracy curves of our method not only surpass those of the baseline methods but also exhibit smaller fluctuations, indicating more consistent task performance.

**Generalization and Forward Transfer.** Forward Transfer (FWT) (Lopez-Paz & Ranzato, 2017) is originally calculated with $A_{t-1,t}$, the test accuracy on task $t$ after the model has learned tasks from 1 to $t-1$, to evaluate how knowledge from previous tasks aids the learning of new tasks. However, before the model sees the data of new task, the classification head cannot correctly map neurons to class IDs, hindering the classification of unseen tasks. Instead, we introduce a new metric called *after one epoch accuracy* (AOA) to assess generalization and forward transfer, which is defined as:

$$\text{AOA} = \frac{1}{T-1} \sum_{i=2}^{T} A_i^{\text{epoch}=1}, \tag{28}$$

where $A_i^{\text{epoch}=1}$ represents the test accuracy on task $i$ after training for one epoch on data $\mathcal{D}_i$ (in the first stage of our framework). Figure 5 shows that our method obtains higher accuracy on nearly all tasks compared to the baselines, highlighting strong inter-task generalizability that accommodates the heterogeneous distribution of new tasks.

**Efficiency.** Despite requiring a second training stage, our method maintains a good balance between efficiency and performance. As shown in Table 4, we compare the training time and GPU memory usage of our method with those of corresponding baselines on the challenging MiniImageNet

*Table 4.* **Efficiency comparison.** While our method requires additional time and GPU memory due to the second training stage, it achieves state-of-the-art performance with competitive efficiency, providing a substantial advantage over the baselines.

| Method | ACC(%) | Training Time (s) | GPU Mem Usage (MB) |
|---|---|---|---|
| GPM | 63.90 | 434.66 | 347.26 |
| GPCNS | 62.85 | 258.13 | 998.41 |
| SGP | 66.99 | 490.84 | 347.26 |
| TRGP | 62.68 | 776.72 | 1854.32 |
| GPM + Ours | 67.62 | 584.14 | 375.72 |
| SGP + Ours | 70.06 | 675.99 | 375.72 |

dataset. The results indicate that our approach enhances performance while requiring significantly less memory than GPCNS and TRGP, and it achieves a shorter training time than TRGP. Although the second stage increases the overall training time, it yields an improvement of more than 3% over the baselines. In addition, updating the Fisher information is computationally efficient, taking only *0.69s* per task – comparable to training for a single epoch – and our method adds no overhead during inference. More results on efficiency are presented in Appendix C.4.

## 5. Conclusion

In this paper, we demonstrate that model merging can improve the overall optimum by identifying a point along the merging path where the cumulative loss is minimized. Additionally, we bridge the gap between prior work and the complexities of task interdependence with theoretical insights based on Bayesian continual learning, offering a closed-form solution for the merging coefficient that balances stability and plasticity. Experiments confirm the effectiveness of our proposed framework BECAME on multiple CL benchmarks. Our findings underscore the potential of model merging to enhance continual learning, providing a flexible and generalizable solution that can be seamlessly incorporated into existing approaches for improvements.

## Impact Statement

Our research adheres to high ethical standards in machine learning, emphasizing transparency, reproducibility, and fairness throughout all experiments. Although our approach demonstrates promising results, it shares the common limitations of computer vision models, particularly in data usage. We use publicly available datasets in accordance with all relevant legal and ethical guidelines, ensuring careful data handling during selection and preprocessing.

To facilitate reproducibility, we provide detailed implementation specifications, including models, datasets, and training setups, in Appendix B. Our code is publicly available at https://github.com/limei0818/BECAME.

## Acknowledgement

This work is supported by the National Nature Science Foundation of China (62176155) and Shanghai Municipal Science and Technology Major Project (2021SHZDZX0102).

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

The appendix is organized as follows:

- Appendix A reviews relevant work on gradient projection-based continual learning methods and model merging.

- Appendix B provides a detailed description of the experimental setup, including supplementary information on the baselines (B.1), dataset statistics (B.2), network architecture (B.3), implementation details (B.4), and evaluation metrics (B.5).

- Appendix C is divided into five parts: Section C.1 presents experimental results on additional datasets, Section C.2 shows the performance with the average anytime accuracy (AAA) metric, Section C.3 compares different merging strategy without the gradient projection training stage, Section C.4 reports the detailed memory usage of GPM-based experiments and efficiency results of NSCL-based experiments, and Section C.5 includes supplementary plots.

## A. Additional Related Work

**Gradient Projection Methods.** The gradient projection method is a significant branch of optimization-based continual learning approaches. Specifically, during training, the method restricts the update direction of parameters to the orthogonal subspace of the input features at each layer of the network. OWM (Zeng et al., 2019) ensures that weight modifications during training for new tasks occur exclusively in directions orthogonal to the input space of previously learned tasks, thereby preventing interference with prior knowledge. Adam-NSCL (Wang et al., 2021b) constrains network parameter updates to lie within the layer-wise null space of input features from previous tasks and incrementally computes the null space based on the uncentered covariance of these features. GPM (Saha et al., 2021) divides the gradient space into two parts and uses learned representations to preserve past knowledge while enabling new task learning through gradient steps that avoid interference with the stored Core Gradient Space. While gradient projection methods exhibit a strong suppression of forgetting, they also limit the model's ability to learn new tasks. Several works aim to improve the model's plasticity based on gradient projection methods. Connector (Lin et al., 2022a) combines Adam-NSCL with regularization-based approaches and employs linear interpolation to achieve a better balance between stability and plasticity in the final model. SGP (Saha & Roy, 2023) enhances new learning by not only allowing gradient updates orthogonal to the feature spaces but also projecting on the feature spaces themselves. TRGP (Lin et al., 2022b) introduces a trust region that selectively reuses relevant old task weights, facilitating effective forward knowledge transfer while minimizing interference with previously learned tasks. GPCNS (Yang et al., 2024) projects current gradients into the common null space of previous tasks' gradients, utilizing both old gradient and feature information.

**Model Merging.** Model merging has emerged as a promising direction for efficiently synthesizing the capabilities of multiple fine-tuned models while reducing data and computational demands. Task Arithmetic (Ilharco et al., 2023) introduces the idea of task vectors that encode task-specific knowledge via the difference between fine-tuned and pre-trained weights. Other approaches such as RegMean (Jin et al., 2023) and FisherMerging (Matena & Raffel, 2022) go beyond simple averaging, leveraging local linear regression or parameter importance scores via the Fisher Information Matrix. Further, methods like TIES-Merging (Yadav et al., 2024) and DARE (Yu et al., 2024) improve merging by resolving parameter interference through pruning and sign alignment strategies. When merging heterogeneous models—those with different architectures—the challenge becomes more complex. Methods such as DAMC (Chen et al., 2024) and FuseLLM (Wan et al., 2024) adopt strategies like parameter decoupling, token alignment, and knowledge distillation. Across these efforts, model merging has been applied to improve generalization (Cha et al., 2021; Arpit et al., 2022), multitask learning (Jin et al., 2023), multimodal fusion (Sung et al., 2023), and continual learning (Marouf et al., 2024).

## B. Experiment Details

### B.1. Baselines

Since our method is proposed based on gradient projection, the primary objective of our experiments is to validate whether adaptive merging can improve the plasticity of gradient projection methods. We mainly compare our method with corresponding gradient projection baselines. Additionally, we include comparisons with the Connector (Lin et al., 2022a), which has the same model capacity and enhances gradient projection by combining it with parameter interpolation and parameter regularization on Adam-NSCL (Wang et al., 2021b). Among all compared methods, OWM (Zeng et al., 2019), Connector, GEM (Lopez-Paz & Ranzato, 2017), A-GEM (Chaudhry et al., 2019a), and FS-DGPM (Wang et al., 2021a) are related to gradient projection. EWC (Kirkpatrick et al., 2017) and SI (Ostapenko et al., 2019) are regularization-based

methods, whereas LwF (Li & Hoiem, 2017), GD-WILD (Lee et al., 2019), and ER-Res (Chaudhry et al., 2019b) are replay-based methods.

### B.2. Datasets

Table 5 and Table 6 summarize the datasets used in our experiments. For GPM-based experiments, the dataset is split into 95% for training and 5% for validation, with no data augmentation applied across all three datasets. In NSCL-based experiments, the entire training dataset is utilized, with data augmentation applied via a random crop with 4-pixel padding and a random horizontal flip.

*Table 5.* Dataset statistics for GPM-based experiments.

|  | **20-Split CIFAR-100** | **10-Split CIFAR-100** | **20-Split MiniImageNet** |
|---|---|---|---|
| Number of Tasks | 20 | 10 | 20 |
| Input Size | $3 \times 32 \times 32$ | $3 \times 32 \times 32$ | $3 \times 84 \times 84$ |
| Classes per Task | 5 | 10 | 5 |
| Training Samples per Task | 4,750 | 4,750 | 2,375 |
| Validation Samples per Task | 250 | 250 | 125 |
| Test Samples per Task | 1,000 | 1,000 | 500 |

*Table 6.* Dataset statistics for NSCL-based experiments.

|  | **20-Split CIFAR-100** | **10-Split CIFAR-100** | **20-Split MiniImageNet** |
|---|---|---|---|
| Number of Tasks | 20 | 10 | 25 |
| Input Size | $3 \times 32 \times 32$ | $3 \times 32 \times 32$ | $3 \times 64 \times 64$ |
| Classes per Task | 5 | 10 | 8 |
| Training Samples per Task | 5,000 | 5,000 | 4,000 |
| Validation Samples per Task | - | - | - |
| Test Samples per Task | 1,000 | 1,000 | 400 |

### B.3. Architecture

**AlexNet-like Architecture**: This architecture mirrors that of Serra et al. (2018), with the addition of batch normalization applied to all layers except the classifier. It features three convolutional layers with filter sizes of 64, 128, and 256, and kernel sizes of $4 \times 4$, $3 \times 3$, and $2 \times 2$, respectively. These are followed by two fully connected layers, each comprising 2048 units. Activation functions throughout are rectified linear units (ReLU), and $2 \times 2$ max-pooling is applied after each convolutional layer. Dropout is employed with rates of 0.2 for the first two layers and 0.5 for the remaining layers.

**Reduced ResNet18 Architecture**: This design is adapted from Lopez-Paz & Ranzato (2017). For the MiniImageNet experiment, the first layer incorporates a convolution with a stride of 2. In both the MiniImageNet and 5-Datasets experiments, the $4 \times 4$ average-pooling layer preceding the classifier is replaced with a $2 \times 2$ average-pooling layer. ReLU activations are used in the hidden layers, while the final layer utilizes softmax with cross-entropy loss.

### B.4. Implementation Details

**GPM-based Experiments.** We validate our method using GPM (Saha et al., 2021) and its three improved variants: TRGP (Lin et al., 2022b), SGP (Saha & Roy, 2023), and GPCNS (Yang et al., 2024). To ensure a fair comparison, the experimental setup is consistent with the corresponding baselines.

The initial learning rate (lr) is reduced by a factor of lr_factor if the validation loss does not improve over lr_patience consecutive epochs. Training terminates when the learning rate falls below lr_min or the maximum number of epochs (n_epochs) is reached. The threshold for selecting orthogonal bases ($\epsilon_{\text{th}}^l$) is initialized as $\epsilon_0$ for the first task and incremented by $\epsilon_\Delta$ for each subsequent task. The scale coefficient ($\alpha$) corresponds to the parameter used in SGP. Hyperparameters for the GPM-based experiments across different baselines are detailed in Table 7. The experiments on both 20-Split CIFAR-100 and 10-Split CIFAR-100 share identical hyperparameters.

*Table 7.* **Hyperparameter settings for each GPM-based baseline.** $^*$ indicates that the parameter value is sourced from the corresponding papers or supplementary materials, while other values are derived from the code.

| Hyperparameter | 20/10-Split CIFAR-100 | | | | 20-Split MiniImageNet | | | |
|---|---|---|---|---|---|---|---|---|
| | **GPM** | **TRGP** | **SGP** | **GPCNS** | **GPM** | **TRGP** | **SGP** | **GPCNS** |
| lr | $0.01^*$ | $0.01^*$ | $0.05^*$ | $0.05^*$ | $0.1^*$ | $0.1^*$ | $0.1^*$ | $0.1^*$ |
| lr_min | $10^{-5}$ | $10^{-5}$ | $5 \times 10^{-5}$ | $5 \times 10^{-5}$ | $10^{-3}$ | $10^{-3}$ | $10^{-3}$ | $10^{-3}$ |
| lr_patience | 6 | 6 | 6 | 6 | 5 | 5 | 5 | 5 |
| lr_factor | 2 | 2 | 2 | 2 | 3 | 3 | 3 | 3 |
| n_epochs | $200^*$ | $200^*$ | $200^*$ | $200^*$ | $10^*$ | $100^*$ | $10^*$ | $200^*$ |
| batchsize | $64^*$ | $64^*$ | $64^*$ | 64 | $10^*$ | $64^*$ | $10^*$ | 64 |
| $\epsilon_0$ | $0.97^*$ | $0.97^*$ | $0.97^*$ | 0.97 | $0.985^*$ | $0.985^*$ | $0.985^*$ | 0.98 |
| $\epsilon_\Delta$ | $3 \times 10^{-3*}$ | $3 \times 10^{-3*}$ | $3 \times 10^{-3*}$ | $3 \times 10^{-3}$ | $3 \times 10^{-4*}$ | $3 \times 10^{-4*}$ | $3 \times 10^{-4*}$ | $10^{-3}$ |
| $\alpha$ | - | - | $10^*$ | $5^*$ | - | - | $1^*$ | $3^*$ |

**NSCL-based Experiments.** Following Adam-NSCL (Wang et al., 2021b), we use the Adam optimizer. The learning rate is set to $10^{-4}$ for the first task and $5 \times 10^{-5}$ for subsequent tasks. To adapt Adam-NSCL to our method, we decay the learning rate by a factor of 0.5 if training loss does not decrease for 3 consecutive epochs, and each stage stop when the condition occurs for the third time. The maximum number of training epochs per stage is 80. The batch size is 32 for 10-Split CIFAR-100, and 16 for both 20-Split CIFAR-100 and 25-Split TinyImageNet.

Parameters in the batch normalization layer are regularized using EWC (Kirkpatrick et al., 2017) with a regularization coefficient of 100. The hyperparameter $a$ in Adam-NSCL is set to 30 for 20-Split CIFAR-100 and 10 for the other two benchmarks. During training on a new task, only the backbone network and the classifier for the current task are updated.

All experiments are performed on a single NVIDIA GeForce RTX 4080 GPU.

### B.5. Metrics

We use two evaluation metrics for the main experimental results: average accuracy (ACC) and backward transfer (BWT) (Lopez-Paz & Ranzato, 2017). ACC measures the overall test performance across all learned tasks, while BWT is quantified by the decrease in test accuracy on previously learned tasks relative to their initial accuracy upon learning. We also use the Intransigence Measure (IM) (Chaudhry et al., 2018) to assess plasticity. IM is calculated as the difference between the upper-bound performance under the multi-task setting and the accuracy achieved immediately after learning each task. These metrics are defined as:

$$\text{ACC} = \frac{1}{T} \sum_{i=1}^{T} A_{T,i}, \quad \text{BWT} = \frac{1}{T-1} \sum_{i=1}^{T-1} A_{T,i} - A_{i,i}, \quad \text{IM} = \sum_{i=1}^{T} A_i^* - A_{i,i}. \quad (29)$$

Here, $T$ denotes the total number of tasks, and $A_{t,i}$ represents the test accuracy of the model after training on task $t$ and evaluated on task $i$, and $A_i^*$ represents the accuracy on task $i$, evaluated using a model trained from random initialization on the combined datasets $\bigcup_{k=1}^{i} \mathcal{D}_k$ .

## C. Additional Results and Plots

### C.1. Experimental Results on Additional Datasets

In addition to CIFAR-100, MiniImageNet, and TinyImageNet, we evaluate our proposed method on the 5-Datasets (Ebrahimi et al., 2020) and CIFAR-100 Superclass (Yoon et al., 2018) benchmarks. The 5-Datasets benchmark is a combination of five different datasets: SVHN, CIFAR-10, not-MNIST, Fashion-MNIST, and MNIST. Each sub-dataset contains 10 classes and is treated as a separate task. The task order significantly influences performance, as learning on the CIFAR-10 sub-dataset is considerably more challenging than on the other four sub-datasets.

For both the 5-Datasets and CIFAR-100 Superclass benchmarks, the network architecture and hyperparameters are identical to those used in the corresponding baselines. The training configuration for the second stage follows that of the first stage in each experiment.

As shown in Table 8, our method consistently improves baseline performance. The performance improvement on the

*Table 8.* **Additional Performance comparison of GPM-based experiments.** We run the experiment on 3 different dataset order. The best ACC is marked in **bold**, and the second-best is underlined. Since GPCNS does not provide code for the 5-Dataset benchmark, the results marked with * are reproduced by us based on its available code for 10-Split CIFAR-100.

| Method | 5-Datasets | | CIFAR-100 Superclass | |
|---|---|---|---|---|
| | ACC(%)↑ | BWT(%)↑ | ACC(%)↑ | BWT(%)↑ |
| GPM (Saha et al., 2021) | $84.70 \pm 3.17$ | $-4.88 \pm 4.37$ | $57.49 \pm 0.65$ | $-1.03 \pm 0.37$ |
| GPM + Ours | $84.89 \pm 3.10$ | $-3.92 \pm 4.61$ | $58.21 \pm 0.22$ | $-0.60 \pm 0.17$ |
| TRGP (Lin et al., 2022b) | $\underline{89.99} \pm 0.58$ | $-0.75 \pm 0.62$ | $57.85 \pm 0.43$ | $-0.93 \pm 0.60$ |
| TRGP + Ours | $\mathbf{90.63} \pm 0.87$ | $-0.80 \pm 0.51$ | $58.07 \pm 0.56$ | $-0.71 \pm 0.30$ |
| SGP (Saha & Roy, 2023) | $83.77 \pm 3.28$ | $-6.25 \pm 5.94$ | $58.24 \pm 0.39$ | $-1.85 \pm 0.54$ |
| SGP + Ours | $84.02 \pm 3.18$ | $-6.24 \pm 6.36$ | $\mathbf{59.03} \pm 0.32$ | $-1.13 \pm 0.28$ |
| GPCNS (Yang et al., 2024) | $75.89 \pm 5.92^*$ | $-17.01 \pm 8.22^*$ | $57.68 \pm 0.84$ | $-3.23 \pm 1.38$ |
| GPCNS + Ours | $76.92 \pm 5.30$ | $-15.51 \pm 7.32$ | $\underline{58.41} \pm 0.13$ | $-2.33 \pm 0.23$ |

5-Datasets benchmark is less than on the 10-Split CIFAR-100 due to two primary factors. First, the key driver of performance improvement on the 5-Datasets benchmark is enhancing the model's learning capability on CIFAR-10, whereas gradients have minimal influence on learning effectiveness for sub-datasets like MNIST. Second, due to the model's poor performance on CIFAR-10, it tends to assign a larger weight to CIFAR-10 during parameter merging, leading to limited enhancement for other tasks.

A similar pattern is observed on the CIFAR-100 Superclass dataset, where new task learning under gradient projection may also converge to local optima.

### C.2. Results with the AAA metric

Despite ACC, we also compute Averaged Anytime Accuracy (AAA) using the formula: $AAA = \sum_{i=1}^{T} AA_i$ with $AA_i = \frac{1}{T} \sum_{j=1}^{t} A_{ij}$. In Tables 9 and 10, we present a comparative analysis of ACC and AAA metrics on different benchmarks side by side. These results confirm that our method continues to yield substantial improvements when evaluated with AAA, with trends closely aligning with ACC.

*Table 9.* **Performance comparison of GPM-based experiments with ACC and AAA metrics.** The best AAA is marked in **bold**, and the second-best is underlined.

| Method | 20-Split CIFAR | | 10-Split CIFAR | | 20-Split MiniImageNet | |
|---|---|---|---|---|---|---|
| | ACC(%)↑ | AAA(%)↑ | ACC(%)↑ | AAA(%)↑ | ACC(%)↑ | AAA(%)↑ |
| GPM (Saha et al., 2021) | 77.34 | 77.31 | 71.81 | 72.08 | 63.90 | 63.02 |
| GPM + Ours | 80.57 | 79.61 | 75.05 | 74.22 | 67.62 | **66.05** |
| TRGP (Lin et al., 2022b) | 81.68 | 79.96 | 75.01 | 74.24 | 62.68 | 62.55 |
| TRGP + Ours | 82.61 | **80.78** | 75.87 | **74.89** | 65.09 | 64.48 |
| SGP (Saha & Roy, 2023) | 80.21 | 78.78 | 74.97 | 73.62 | 66.99 | 61.97 |
| SGP + Ours | 81.94 | 79.44 | 76.74 | 74.82 | 70.06 | 64.98 |
| GPCNS (Yang et al., 2024) | 78.63 | 76.22 | 71.84 | 71.50 | 62.85 | 60.53 |
| GPCNS + Ours | 80.87 | 78.08 | 73.89 | 72.94 | 64.79 | 61.53 |

### C.3. Merging Strategy Compatison without GP

To isolate the impact of the gradient projection methods, we directly evaluate different model merging strategies in CL setting. Except lack of the first training stage, all other training configurations remain the same as in the "Merging Strategy" part of Section 4.3.

As shown in Table 11, our adaptive merging method achieves the highest ACC on both 10-Split CIFAR100 and 20-Split MiniImageNet, outperforming fixed-coefficient methods like $1/t$ and CoMA. Furthermore, our method maintains a relatively

*Table 10.* **Performance comparison of NSCL-based experiments with ACC and AAA metrics.** The best AAA is marked in **bold**.

| Method | 20-Split CIFAR | | 10-Split CIFAR | | 25-Split TinyImageNet | |
|---|---|---|---|---|---|---|
| | ACC(%)↑ | AAA(%)↑ | ACC(%)↑ | AAA(%)↑ | ACC(%)↑ | AAA(%)↑ |
| Adam-NSCL (Wang et al., 2021b) | 75.66 | 75.43 | 72.91 | 74.77 | 58.77 | 60.21 |
| Adam + Ours | 81.88 | **81.25** | 81.66 | **81.94** | 66.49 | **66.65** |

high BWT, indicating better retention of previous tasks. These results demonstrate the robustness and effectiveness of our adaptive merging strategy compared to other merging strategies.

*Table 11.* **Performance comparison without gradient projection of different merging strategies on 10-Split CIFAR-100 and 20-Split MiniImageNet based on GPM.**

| Method | 10-Split CIFAR100 | | 20-Split MiniImageNet | |
|---|---|---|---|---|
| | ACC(%)↑ | BWT(%)↑ | ACC(%)↑ | BWT(%)↑ |
| Finetune | 57.98 | -20.19 | 57.06 | -10.93 |
| $1/t$ (Lee et al., 2017) | 60.69 | -1.44 | 47.54 | 0.91 |
| CoMA (Marouf et al., 2024) | 64.42 | -9.79 | 60.99 | -5.00 |
| CoFiMA (Marouf et al., 2024) | 61.17 | -0.38 | 62.26 | -1.40 |
| **Ours** | **65.03** | -1.26 | **64.75** | -2.11 |

## C.4. Additional Efficiency Results

**Detailed memory usage on GPM-based experiments.** To provide a detailed comparison, we divide the total memory into three parts: model parameters, the additional memory for variables needed for training future tasks, and the temporary memory required only during training. We further divide the additional memory based on training stage for analysis. Table 12 shows that:

- The second training stage introduces only a modest storage overhead. As shown in the 5th and 6th rows of the table, the only additional memory for the second training stage is the precision matrix, which is comparable in size to the model parameters and determined solely by the model architecture rather than by the baseline method.

- The increment of the additional memory in the first training stage is a direct result of improved overall performance, which follows the inherent feature of the gradient projection methods. After performing our model merging, the model's capability of feature extraction is enhanced so that the dimension of output feature space expands and more base vectors need to be stored for future task training.

- The memory overhead of our method is competitive to GPCNS and TRGP, which require more storage than our method even though they only have one training stage.

**Efficiency analysis on NSCL-based experiments.** We evaluated the training time and GPU memory usage in NSCL-based experiments, as shown in Table 13. In this table, Adam-NSCL serves as the baseline for our method, while Connector represents an alternative approach designed to improve upon Adam-NSCL. Our method demonstrates a significant performance improvement over Adam-NSCL. Furthermore, it not only surpasses Connector in performance but also requires significantly less training time and memory.

## C.5. Additional Plots

**Plasticity-Stability Trade-off.** Figure 6 displays the BWT and IM from the GPM-based experiments, presented as a single plot for each dataset. The symbol + denotes our method applied to the corresponding baseline. Compared to GPM, all other three baselines – including GPCNS, TRGP, and SGP – show improved plasticity, albeit with some potential reduction

*Table 12.* **Detailed memory usage on GPM-based experiments.**

| Method | ACC (%) | Model Parameters (MB) | Additional Memory (MB) | | Temporary Memory (MB) |
|---|---|---|---|---|---|
| | | | 1st Stage | 2nd Stage | |
| GPM | 63.90 | 4.74 | 37.51 | – | 305.01 |
| GPCNS | 62.85 | 4.74 | 584.26 | – | 409.41 |
| SGP | 66.99 | 4.74 | 53.47 | – | 289.05 |
| TRGP | 62.68 | 73.23 | 1359.18 | – | 421.91 |
| GPM+Ours | 67.62 | 4.74 | 55.88 | **4.73** | 310.37 |
| SGP+Ours | 70.69 | 4.74 | 60.49 | **4.73** | 305.76 |

*Table 13.* **Efficiency comparison with NSCL-based methods on 25-Split Tiny ImageNet.**

| Method | ACC (%) | Training Time (h) | GPU Mem Usage (MB) |
|---|---|---|---|
| Adam-NSCL (Wang et al., 2021b) | 58.57 | 5.22 | 1728.71 |
| Connector (Lin et al., 2022a) | 64.61 | 8.53 | 1766.66 |
| NSCL+Ours | 66.49 | 6.30 | 1748.77 |

in stability. Notably, our method further enhances plasticity relative to the baselines, contributing to improved overall performance.

Figure 7 presents the BWT and IM from the NSCL-based experiments. Our method achieves significant improvements in IM.

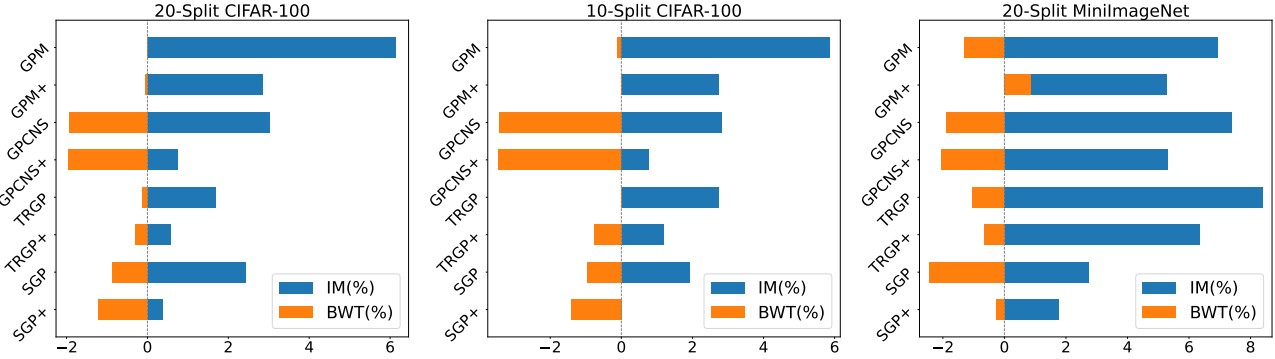

*Figure 6.* **BWT and IM for GPM-based Experiments.** Each dataset is represented in a single plot. The symbol + indicates our method applied to the corresponding baseline.

**Balance Across Tasks.** Figure 8 depicts the final accuracy across all tasks for the GPM-based experiments, with each column representing a specific dataset. Our method not only surpasses the baselines in ACC but also achieves a low standard deviation. Similarly, Figure 9 presents the results for the NSCL-based experiments.

**Generalization and Forward Transfer.** Figure 10 and 11 illustrate the accuracy after one epoch across all tasks for GPM-based and NSCL-based experiments respectively.

**Coefficient *vs*. Loss.** We sample the merge coefficient from 0 to 1 with a step size of 0.05 to analyze the impact of different coefficients on the cumulative training loss of tasks learned. Figures 12 and 13 illustrate the results of experiments based on GPM and Adam-NSCL, respectively, on the 10-Split CIFAR-100 dataset. The vertical dashed lines in the figures indicate the coefficients computed using our method, which identify a point with the minimum cumulative loss for different tasks. A comparison of Figures A and B reveals that the optimal merge coefficients differ across models on the same dataset. Our method adaptively identifies the optimal merge coefficient.

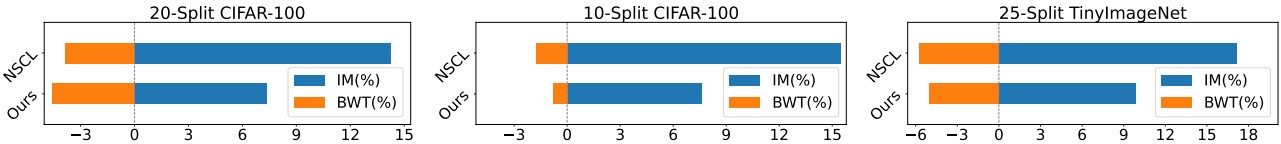

*Figure 7.* **The BWT and IM from the NSCL-based experiments.**

*Figure 8.* **Final accuracy across all tasks in the GPM-based experiments.**

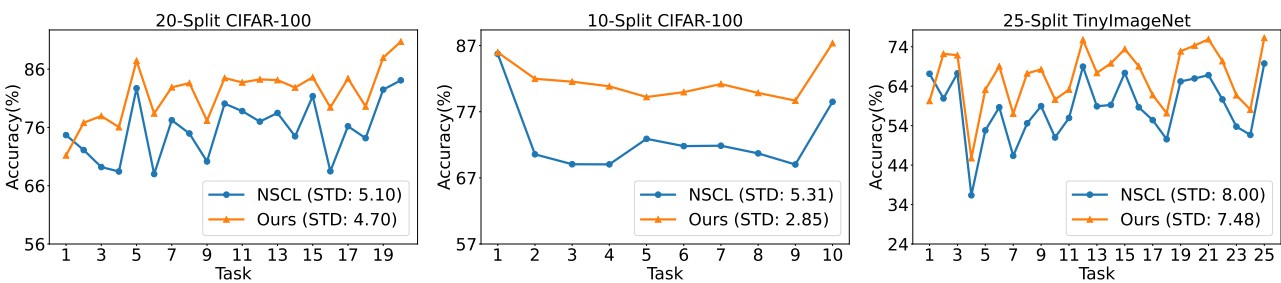

*Figure 9.* **Final accuracy across all tasks in the NSCL-based experiments.**

*Figure 10.* **After one epoch accuracy in GPM-based experiments.**

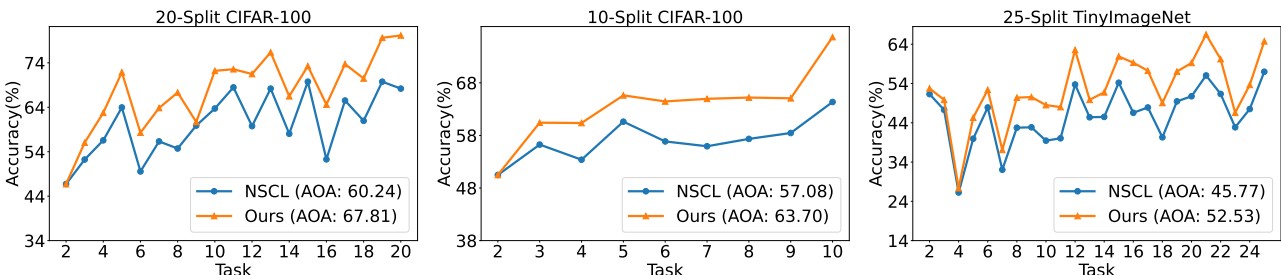

*Figure 11.* **After one epoch accuracy in NSCL-based experiments.**

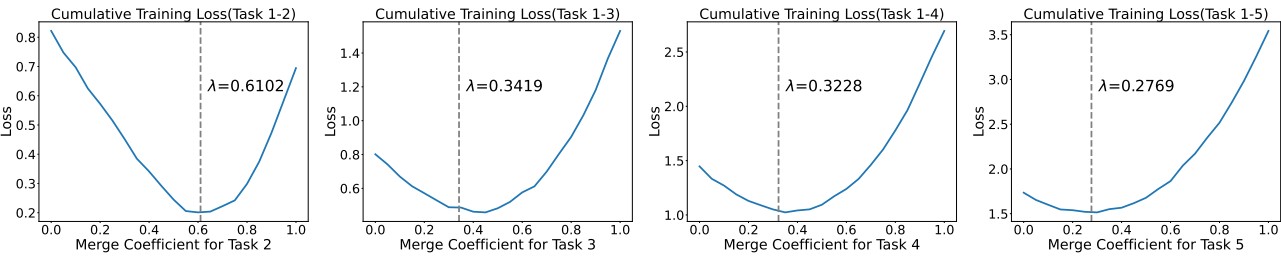

*Figure 12.* **Cumulative training loss versus different merge coefficient on 10-Split CIFAR-100 dataset in NSCL-based experiments.** The dashed line denotes the merge coefficient calculated by our method.

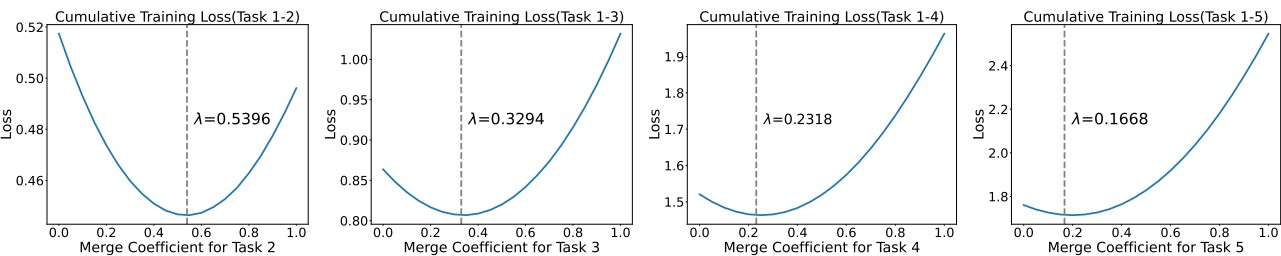

*Figure 13.* **Cumulative training loss versus different merge coefficient on 10-Split CIFAR-100 dataset using GPM as baseline.**

