# OpenReview forum: "BECAME: Bayesian Continual Learning with Adaptive Model Merging"
_ICML.cc/2025/Conference — ICML 2025 poster_

### Official Review · Reviewer_EVng · 2025-03-04

**Overall Recommendation:** 2

**Summary:**

The paper proposes BECAME, a Bayesian continual learning framework that adaptively merges task-specific models to balance stability and plasticity. Key contributions include:

* A closed-form solution for merging coefficients derived via Bayesian principles, proving that merging models along a linear path can achieve a lower cumulative loss than individual task-optimized models.

* A two-stage training paradigm combining gradient projection (for stability) and unconstrained optimization (for plasticity), followed by adaptive model merging.

* Some theoretical analysis is provided.

* Extensive experiments on CL benchmarks demonstrate state-of-the-art performance.

**Claims And Evidence:**

Yes, the main claims are supported by evidence.

**Essential References Not Discussed:**

This submission discussed model merging, but one closely related paper about adaptive model merging [1] has not been discussed.

Reference:

[1] AdaMerging: Adaptive Model Merging for Multi-Task Learning, ICLR 2024.

**Experimental Designs Or Analyses:**

The experimental designs and analysis are sound and valid.

**Methods And Evaluation Criteria:**

Yes, this paper evaluates different CL approaches with overall accuracy and backward transfer, which are commonly used evaluation metrics.

**Other Comments Or Suggestions:**

N/A

**Other Strengths And Weaknesses:**

**Strength**:
* The paper introduces an approach to continual learning (CL) by leveraging Bayesian principles to derive a closed-form solution for adaptive model merging.

* The theoretical analysis and the derivation of the optimal merging coefficient are reasonable. The authors provide a clear mathematical framework to explain why merging models along a linear path can lead to better minima for cumulative loss across tasks.



**Weakness**:
* **Computational Overhead**: While merging itself is efficient, the two-stage training (gradient projection + unconstrained training) doubles training time compared to single-stage baselines.

* **Memory Overhead**: The author only compares the GPU memory consumption. I think this comparison is not fair, it would be better to compare the memory cost of storing additional set of model parameters and others.

**Questions For Authors:**

N/A

**Relation To Broader Scientific Literature:**

This paper improves continual learning through adaptive model merging. The model merging have been discussed in the related works.

**Theoretical Claims:**

To the best of my knowledge, the proof looks correct.

---

> ### Author Rebuttal · Authors · 2025-04-01
>
> We appreciate your time and highlighting these points. Below, we address each concern in detail.
> ## Reference
> We acknowledge the significance of AdaMerging[1] in model merging and promise to cite it to recognize its contribution. Additionally, we will expand the appendix to **provide an comprehensive discussion on model merging**.
>
> After a careful review of the AdaMerging paper, we identify several key distinctions between their approach and ours. While both methods address adaptive model merging, AdaMerging operates within a multi-task learning framework where tasks are learned **concurrently**. In contrast, our work focuses on continual learning, where tasks are learned **sequentially**. Our merging strategy is specifically designed to consider the temporal dependencies between tasks.
>
> Moreover, AdaMerging iteratively optimizes the merging coefficient at test time using entropy minimization on unlabeled data. In our approach, the coefficient is determined during training by a closed-form solution.
> ## W1
> While our two-stage training approach introduces additional computational costs, our results show that the overall training time **remains well below double** **that of baselines**, and the performance improvements justify the extra overhead. For instance, as shown in Table 4, GPM attains 63.90% ACC in 434.66s, whereas our method achieves 67.62% in 584.14s. The second stage adds **less than 35%** to the duration of the first stage, making our approach highly competitive compared to single-stage methods like TRGP, which requires considerably more training time (776.72s) due to complex operations.
>
> The reasons for this efficiency are as follows:
> - The unconstrained training in second stage incurs a lower per-epoch cost (0.52s) than gradient projection (0.85s) and typically converges in fewer epochs when early stopping is applied.
> - In contrast to the gradient projection, which spends additional time computing the subspace for each task, the second stage avoids this overhead entirely.
>
> ## W2
> We would like to highlight that **our method does not significantly increase storage requirements.**
>
> Below, we first present the experimental results and then analyze why the BECAME framework is memory-efficient.
>
> To provide a detailed comparison, we divide the total memory into three parts: model, the additional memory for variables needed for training future tasks and the temporary memory required only during training. We further divide the additional memory based on training stage for analysis.
> The results are as follows:
>
> |          | ACC (%) | Total Memory (MB) | Model Parameters (MB) | Additional Memory in 1st Stage (MB) | Additional Memory in 2nd Stage (MB) | Temporary Memory (MB) |
> | -------- | ------- | ----------------- | --------------------- | ----------------------------------- | ----------------------------------- | --------------------- |
> | GPM      | 63.90   | 347.26            | 4.74                  | 37.51                               | –                                   | 305.01                |
> | GPCNS    | 62.85   | 998.41            | 4.74                  | 584.26                              | –                                   | 409.41                |
> | SGP      | 66.99   | 347.26            | 4.74                  | 53.47                               | –                                   | 289.05                |
> | TRGP     | 62.68   | 1854.32           | 73.23                 | 1359.18                             | –                                   | 421.91                |
> | GPM+Ours | 67.62   | 375.72            | 4.74                  | 55.88                               | **4.73**                            | 310.37                |
> | SGP+Ours | 70.69   | 375.72            | 4.74                  | 60.49                               | **4.73**                            | 305.76                |
> - **The second training stage introduces only a modest storage overhead.** As shown in the 5th and 6th rows of the table, the only additional memory for the second training stage is the precision matrix, which is comparable in size to the model parameters and determined solely by the model architecture rather than by the baseline method.
> - The increment of the additional memory in the first training stage is a direct result of improved overall performance, which follows the inherent feature of the gradient projection methods. After performing our model merging, the model's capability of feature extraction is enhanced so that the dimension of output feature space expand and more base vectors need to be stored for future task training.
> - The memory overhead of our method is competitive to GPCNS and TRGP, which required more storage than our method even they only have one training stage.
>
>
> We hope these detailed responses address your concerns and further enhance the robustness of our approach.

---

### Official Review · Reviewer_mavm · 2025-03-10

**Overall Recommendation:** 4

**Summary:**

The paper presents BECAME, a Bayesian Continual Learning framework designed to address the stability-plasticity dilemma in continual learning. The method combines gradient projection methods with model merging to balance retaining prior knowledge (stability) and learning new tasks (plasticity). The key contribution is deriving a closed-form solution for optimal merging coefficients using Bayesian principles. Besides, the proposed method is simple and compatible with various gradient projection CL methods. The proposed method is validated by extensive experiments on multiple benchmarks. It achieves superior performance.

## update after rebuttal
The authors have addressed my concerns regarding BWT.  However, the AAA results are often only comparable to or worse than Acc. These results contradict the trends in existing works (AAA typically exceeds Acc by a significant margin). This is a remaining minor concern.

Overall, I appreciate that the proposed method is simple yet supported by theoretical inference of the optimal combination coefficient. Therefore, I am inclined towards the acceptance of the paper.

**Claims And Evidence:**

Yes. This is one of its strengths. For example, it supports the motivation of adaptive model merging using illustration and empirical evidence as shown in Figures 1 and 2.

**Essential References Not Discussed:**

Essential references are discussed.

**Experimental Designs Or Analyses:**

The experimental designs are good. It sufficiently supports the effectiveness of the proposed method. A potential improvement is to use AAA as a metric for evaluating CL methods.

**Methods And Evaluation Criteria:**

The method is simple but makes sense. It is supported by empirical evidence and theoretical reasoning that adaptive model merging achieves better performance.

The benchmarks and metrics are sufficient to evaluate the proposed methods. But this paper does not report the Averaged Anytime Accuracy (AAA) performance, which is a widely used metric for studying continual learning.

**Other Comments Or Suggestions:**

1. Add AAA as a metric to evaluate CL methods.
2. Fix the minor issue "the log prior log p(θ) is not related to the optimization as it is a constant under a certain initialization" in the right column lines 181-182. The statement holds in the specific case of a uniform prior, instead of under a certain initialization. It does not apply to MAP estimation in general.

**Other Strengths And Weaknesses:**

Strengths:
1. The proposed method is simple but novel. And its effectiveness is demonstrated through theory reasoning and empirical results.
2. The paper is well-written and easy to follow. It provides a clear illustration of motivation as in Figure 1 and Figure 2.
3. Extensive experimental results to validate its effectiveness.

Weakness:
1. Combining with the proposed method may produce worse BWT in some cases, but not in most.

**Questions For Authors:**

1. Can the proposed method be extended to class incremental settings?

**Relation To Broader Scientific Literature:**

This paper proposed a novel CL method, which is simple yet effective. It can be applied to different applications.

**Theoretical Claims:**

I have checked the theory part. Most of them are correct. A minor issue might be the statement that "the log prior log p(θ) is not related to the optimization as it is a constant under a certain initialization" in the right column lines 181-182. The statement holds in the specific case of a uniform prior, instead of under a certain initialization. It does not apply to MAP estimation in general.

---

> ### Author Rebuttal · Authors · 2025-04-01
>
> We appreciate your careful review and the constructive suggestions. Your recognition of this work is truly encouraging. Here we present our detailed clarifications and updates, which we believe further enhance our paper.
> ## Evaluation&C1
> In response to your suggestion, we compute Averaged Anytime Accuracy (AAA) using the formula: $AAA = \sum_{i=1}^T AA_i$ with $AA_i = \frac{1}{i} \sum_{j=1}^{i} A_{i,j}$. The results are shown in the tables below:
>
> |            | 20-S CIFAR |           | 10-S CIFAR |           | 20-S MiniImageNet |           |
> | ---------- | ---------- | --------- | ---------- | --------- | ----------------- | --------- |
> |            | AAA        | ACC       | AAA        | ACC       | AAA               | ACC       |
> | GPM        | 77.31      | 77.34     | 72.08      | 71.81     | 63.02             | 63.90     |
> | GPM+Ours   | 79.61      | 80.57     | 74.22      | 75.05     | **66.05**         | 67.62     |
> | TRGP       | 79.96      | 81.68     | 74.24      | 75.01     | 62.55             | 62.68     |
> | TRGP+Ours  | **80.78**  | **82.61** | **74.89**  | 75.87     | 64.48             | 65.09     |
> | SGP        | 78.78      | 80.21     | 73.62      | 74.97     | 61.97             | 66.99     |
> | SGP+Ours   | 79.44      | 81.94     | 74.82      | **76.74** | 64.98             | **70.06** |
> | GPCNS      | 76.22      | 78.63     | 71.50      | 71.84     | 60.53             | 62.85     |
> | GPCNS+Ours | 78.08      | 80.87     | 72.94      | 73.89     | 61.53             | 64.79     |
>
> |           | 20-S CIFAR |           | 10-S CIFAR |           | 25-S TinyImageNet |           |
> | --------- | ---------- | --------- | ---------- | --------- | ----------------- | --------- |
> |           | AAA        | ACC       | AAA        | ACC       | AAA               | ACC       |
> | Adam      | 75.43      | 75.66     | 74.77      | 72.91     | 60.21             | 58.77     |
> | Adam+Ours | **81.25**  | **81.88** | **81.94**  | **81.66** | **66.65**         | **66.49** |
>
> These results confirm that our method continues to **yield substantial improvements when evaluated with AAA**, with trends closely aligning with ACC. Due to space limit, we will include these results in the appendix.
>
> ## Theory&C2
> Thank you for your suggestion. We will revise the statement at lines 181–182 to clarify that the log prior remains constant only under a uniform prior.
> Additionally, we have verified that the parameter prior is predefined and that this adjustment does not affect subsequent derivations or results.
> ## W1
> The observed degradation in the Backward Transfer (BWT) metric in some cases reflects a deliberate trade-off favoring enhanced plasticity and overall performance.
>
> - **Sensitivity of BWT.** BWT of task t is determined by its immediate post-training performance ($A_{t,t}$) and its final performance after learning all tasks ($A_{T,t}$). In some instances, a higher $A_{t,t}$ may lead to a lower BWT even when overall ACC improves.
> - **Stability-Plasticity trade-off.** Our approach intentionally prioritizes increased plasticity to boost performance on new tasks, which may slightly reduce stability (as measured by BWT). However, the overall accuracy benefits from this trade-off.
>
> We emphasize that our method does not compromise the performance of earlier tasks beyond an unacceptable level; rather, it strategically **balances the stability-plasticity trade-off** to achieve better overall performance.
> ## Q1
> Yes, our method is **equally applicable to class-incremental learning** (CIL). The theoretical framework and model merging mechanism are independent of task- or class-incremental learning. Since our approach operates directly on model parameters, it is effective in both scenarios.
> To validate this, we conducted a simple experiment by testing without task IDs, following the standard setup for CIL.
>
> |          | 20-S CIFAR | 10-S CIFAR | 20-S MiniImageNet |
> | -------- | ---------- | ---------- | ----------------- |
> | GPM      | 24.73      | 30.73      | 20.98             |
> | GPM+Ours | **27.12**  | **37.04**  | **27.04**         |
>
> |           | 20-S CIFAR | 10-S CIFAR | 25-S TinyImageNet |
> | --------- | ---------- | ---------- | ----------------- |
> | Adam      | 12.29      | 18.09      | 6.03              |
> | Adam+Ours | **17.84**  | **31.79**  | **13.37**         |
>
> The results indicate that our method also **improves accuracy in the CIL setting**. The improvement may be attributed to the enhanced balance of performance across tasks achieved through merging (see Line 408). We are delighted to investigate this aspect more in future work.
>
>
> We hope these detailed responses clarify our revisions and validate the robustness of our approach. We sincerely appreciate your valuable feedback.

---

### Official Review · Reviewer_H5hN · 2025-03-13

**Overall Recommendation:** 3

**Summary:**

This paper introduces a novel framework called BECAME to address a crucial problem in continual learning, i.e., retaining prior knowledge while learning new tasks to achieve stability and plasticity. From the perspective of Bayes continual learning, BECAME develops a novel merging mechanism to bridge the gap between prior work and the complexities of task interdependence, providing a theoretical demonstration of the stability-plasticity trade-off. Specifically, the optimal merging coefficient for two successive models can be derived via a closed-form solution. Extensive experiments demonstrate the superior performance of BECAME, suggesting its effectiveness in finding an optimal merging model that maximizes overall performance.

**Claims And Evidence:**

The claims in the submission are supported by clear and convincing evidence.

**Essential References Not Discussed:**

This paper has fully discussed the related work, and there is no more essential literature to my knowledge.

**Experimental Designs Or Analyses:**

The experiments, conducted on several widely used benchmarks, exhibit outstanding performance. The experimental designs and analyses are sound and make sense.

**Methods And Evaluation Criteria:**

The proposed method makes sense for the problem of continual learning.

**Other Comments Or Suggestions:**

This paper is well written and I didn't notice any typos. My advice is to provide a more detailed discussion of the related work about model merging in the appendix like gradient projection. Moreover, more classic methods about Gaussian mixture model and multivariate mixtures could be cited.

**Other Strengths And Weaknesses:**

Strengths:

1. The method provides theoretical insights into model merging to achieve optimal performance in continual learning. The optimization objective has been proven to be convex and the closed-form solution can be efficiently obtained.
1. The paper is well written and easy to follow; extensive experiments have validated the effectiveness of the merging strategy.

Weakness:

1. In the first stage of BECAME, the method incorporates the GP method to enhance stability in the previous task. However, the MAP parameter should be $\theta_{t-1}^*$ in section 3.3. To fully validate the effectiveness of the merging strategy, it is important to conduct ablation studies starting from the previously obtained parameter $\theta_{t-1}^*$, as well as to compare it with other model merging strategies.

**Questions For Authors:**

1. It is a pity that the article does not fully incorporate GP into the theoretical framework. However, the authors claim that $\theta_{t-1}^*$ can be substituted with $\theta_t^{GP}$, this process seems more like an experimental conclusion. Since the MAP parameter is supposed to be $\theta_{t-1}^*$ rather than $\theta_t^{GP}$ in the derivation, I would prefer a general framework that theoretically analyzes the result in Line 267.
2. This paper first follows the Streaming Bayes theorem to derive the MAP problem in Eq. 7, then utilizes Laplace approximation to deal with the previous posterior $p(\theta|D_{1:t-1})$. The obtained formulation in Eq. 10 seems to be equivalent to former work. Even if we treat Eq. 10 as a convex problem, the optimal result does not necessarily lie on the line between $\theta_{t-1}^*$ and $\hat\theta_t$, What is the advantage of linear merging rather than jointly minimizing Eq. 10 by treating the latter part as a regularization term? If it does, can we regard Eq. 11 as a sub-problem of Eq. 10 with the constraint that $\theta$ is a linear combination of $\theta_{t-1}^*$ and $\hat\theta_t$? Does the closed-form solution avoid any explicit parameter calculations compared with other regularization-based methods, given that $\hat\theta_t$ still needs to be optimized? What if the updating direction of regularization-based method is restricted to lie between $\theta_{t-1}^*$ and $\hat{\theta}_t$, can we still obtain a similar optimal result? Why or why not?

**Relation To Broader Scientific Literature:**

This paper addresses the problem of continual learning from the perspective of Bayes learning. Although the fundamental objective problem is based on prior literature, it presents a novel approach to merging models and achieves stability and plasticity.. Specifically, the linear combination weights can be obtained through a closed-form solution. I believe this research can inspire Bayes continual learning and model merging.

**Theoretical Claims:**

Yes, I have checked the correctness of the proofs and theoretical claims in the main body, and I have several concerns:

1. The problem definition missed some important hypotheses. For instance, the data are supposed to be iid; otherwise, Eqs. 1 and 6 can not be established.
2. The proof of Lemma 3.1 seems to default to the fact that the loss function is convex (line 182), but it has not been stressed out in the preliminary. However, under this circumstance, Lemma 3.1 seems to be a little trivial, as the summation of two convex function ($L_{1:t-1}$, $L_{t}$) must be convex, such that the new minimum is between $\theta^*_{t-1}$ and $\hat{\theta}_t$, Eq. 2 can then be easily obtained by the nature of convex function.

---

> ### Author Rebuttal · Authors · 2025-04-01
>
> We sincerely appreciate your thorough review and the insightful comments on our theoretical framework and experimental validations. Below, we detail our clarifications and additional proof.
> # Theory
> 1. We agree that the assumption of independence across tasks is necessary for Eq. 6 and will fix it, while Eq. 1 is derived by definition and imposes no such constraint.
> 2. Our proof of Lemma 3.1 utilizes fixed endpoints to show the existence of a merged model with a lower cumulative loss. Importantly, this proof **does not assume that the** **loss function** **is globally convex** with respect to the model parameters.
> # W1
> We conduct additional ablation studies in response to your advice. And the results indicate our method **consistently outperforms alternative merging strategies**.
>
> **Please refer to the Experiment part of the response to reviewer Nkj6.** We will include them in appendix to further support our method.
>
> # C1
>
> We promise to include a detailed discussion on model merging in appendix in our revised vision, as well as classical literature on Gaussian mixture models and multivariate mixtures.
>
> # Q1
> Thank you for the suggestion, here we supplement a theoretical analysis on why $\Delta\theta$ in Eq. 17 can be substituted by $\hat{\theta}_t-\theta_t^{GP}$.
>
> The key point is to prove that Eq. 10 holds when $\theta_{t-1}^*$ is replaced by $\theta_t^{GP}$.
>
> Given $\Lambda_{t-1}$​ is symmetric and semi-positive definite, as it is the Hessian of the negative log posterior, we have
> $$
> (\theta-\theta_t^{GP})^{\top}\Lambda_{t-1}(\theta-\theta_t^{GP})=(\theta-\theta_{t-1}^*)^{\top} \Lambda_{t-1}(\theta-\theta_{t-1}^*)+2(\theta-\theta_{t-1}^*)^{\top}\Lambda_{t-1}(\theta_t^{GP}-\theta_{t-1}^*)+(\theta_t^{GP}-\theta_{t-1}^*)^{\top}\Lambda_{t-1}(\theta_t^{GP}-\theta_{t-1}^*).
> $$
> Then we only need to prove $\Lambda_{t-1}(\theta_t^{GP}-\theta_{t-1}^*)=0 \ (1)$. We define $d=\theta_t^{GP}-\theta_{t-1}^*$.
>
> $\Lambda_{t-1}$ can be decomposed as $\Lambda_{t-1}=Q^{\top}AQ$, as $A$ is a diagonal matrix of the eigenvalues.
>
> To prove (1), we do a second-order Taylor expansion of $L_{1: t-1}(\theta_t^{GP})$ around $\theta_{t-1}^*$, yielding: $d\Lambda_{t-1}d=dQ^{\top}AQd=0$, since $L_{1: t-1}(\theta_t^{GP})\approx L_{1: t-1}(\theta_{1-t}^*)$ and $\theta_{t-1}^*$ is an optimum.
>
> Given that $A_{ii}\geq 0$, it follows each element of $Q$ must be 0, leading to $\Lambda_{t-1}d=Q^{\top}AQd=0$​​​​.
>
> Due to the character limit, we will provide a detailed proof in our revised version.
>
> # Q2
> 1. Our theory does not require Eq. 10 to be convex. We only claim that when merging model from$θ_{t−1}^*$ to $\hatθ_t$, the total loss in Eq.11 **is convex with respect to** $\lambda$ (Line 236).
> 2. We treat $\hatθ_t$ as a scalar for the following reasons:
> - When $\lambda$ is scalar, our analysis allows us to readily show that $\tilde L_{1:t}(\lambda)$ is convex .
> - From our proof in Q1, when $\lambda$ is a vector, $\Lambda_{t-1}\lambda(\theta_t^{GP}-\theta_{t-1}^*)$ may not be 0, preventing the replacement of $\theta_{t-1}^*$ with $\theta_t^{GP}$ in our BECAME framework.
> - There exists a linear connector from $\theta_A$ to $\theta_B$ when they are trained sequentially [1], as also indicated in the right column of Line 146.
> - Our ablation studies and Table 3 have demonstrated that per-parameter merging does not necessarily yield better results.
> 3. We also provide additional experiments and analysis to validate that optimizing Eq. 10 via model merging is more efficient than regularization.
>
> | |10-S CIFAR ACC (%)|BWT (%)|Train Time (s)|20-S MiniImageNet ACC (%)|BWT (%)|Train Time (s)|
> |---|---|---|---|----|----|---|
> |Regularization|59.76|-19.58|218.9|58.90|-12.41|612.5|
> |Limited Reg|62.40|-14.28|341.98|58.24|-11.49|931.1|
> |Ours (Merge)|65.03|-1.26|108.7|64.71|-2.61|232.7|
> |GPM+Regularization|71.99|-7.72|362.5|64.69|-4.19|781.2|
> |GPM+Limited Reg|72.35|-5.34|527.2|64.94|-3.69|1083.1|
> |**GPM+Ours**|**75.05**|**0.02**|274.2|**67.62**|**0.87**|584.14|
>
> Now we present the analysis to further elaborate:
> - Adding regularization to the loss function slows down training. The additional time grows as the parameter dimension increases, whereas model merging without regularization leads to faster training.
> - Regularization is added to prevent the loss of old tasks from increasing. For a specific $\theta_i$ in $\theta$, optimization for the new task follows the loss based on the network architecture, while for old tasks it is constrained by the second-order norms. This explains why the results of regularization differ from those of model merging, even with constrained direction. **Our approach naturally avoids this conflict** and consistently achieves the best results.
> - Due to the aforementioned imbalance, weight of the regularization term should be carefully tuned to get optimal performance, while our model merging **does not require such manual tuning**.
>
> [1] Linear mode connectivity and the lottery ticket hypothesis. ICML 2020

---

### Official Review · Reviewer_Nkj6 · 2025-03-13

**Overall Recommendation:** 2

**Summary:**

The paper proposes a method for continual learning based on updating the parameters for old tasks under an approach with limited plasticity, e.g. gradient projection methods, and then merging with parameters trained more freely for the new tasks. The paper proposes an approach for determining the merging coefficient in closed form based on the curvature around the minima. The experiments show improved performance over a range of recent gradient-projection based methods and widely known baselines from the literature on a vision classification benchmarks (split CIFAR100, Split TinyImagenet) with AlexNet and ResNet based architectures.

**Claims And Evidence:**

Claims closed form optimal solution for merging coefficient, but this is not accurate and quite misleading to readers who do not go through the full paper. The derivation is based on an approximation of the objective, hence the closed form expression is not guaranteed to be optimal for the true objective. This needs to be clearly acknowledged throughout the paper.

**Essential References Not Discussed:**

n/a

**Experimental Designs Or Analyses:**

The baselines are problematic. The experiments mainly demonstrate that the proposed merging approach can be combined with different gradient projection methods. However, the core technical contribution is not compared to alternative approaches for merging and this comparison is essential to evaluating a core technical contribution of the paper (the merging approach).

**Methods And Evaluation Criteria:**

Yes.

**Other Comments Or Suggestions:**

I hope the authors do not take the score as an overly harsh judgement of their work. It is unfortunately the only "reject" option that isn't borderline. The overall merging pipeline seems valuable, however I think the theoretical sections (3.3 in particular) and experiments (other merging baselines) will have to be reworked significantly and render this work better suited for resubmission rather than revision.

**Other Strengths And Weaknesses:**

Strengths:
* The paper is clear
* The approach is well-motivated and the overall pipeline makes sense
* The merging approach improves results for a range of gradient projection based continual learning methods

Weaknesses:
* It is not clear to me that the proposes merging technique is not just Fisher merging with a single coefficient.
* The merging approach is not compared to alternatives, which I find problematic given that it is the core technical contribution of the paper. It is neat that it works well for a range of gradient projection methods, however I find this less relevant than substantiating the claim that the merging is optimal. In particular, Fisher merging (assuming I am correct in suspecting the close relationship) tends to be a relatively weak baseline, at least in the LLM merging literature. So I would suspect that there likely is a better-performing alternative off the shelf.
* The paper consistently claims the theoretical optimality of its merging coefficient, which appears to be incorrect given that the closed form solution is based on an approximation of the objective.

**Questions For Authors:**

* Could you comment on the relationship between your merging technique and Fisher merging?
* If you believe that your merging coefficient is actually optimal, could you explain why? I appreciate the preceding discussion in 3.2 about the absence of loss barriers. But (a) this seems to be a conjecture rather than theoretically guaranteed and (b) even with an absence of barriers, the approximation of the objective seems to break any guarantees to me.

I just want to state explicitly in advance that I may not end up raising my score if these questions are answered, as my core concern is the lack of baselines for the proposed merging method.

**Relation To Broader Scientific Literature:**

The paper explains its motivations clearly in improving on previous gradient projection methods for continual learning. Merging methods are briefly discussed in the related work section. The method seems very closely related to Fisher merging, with the difference being that the present work uses a scalar rather than a per-parameter merging coefficient. Unfortunately, this relationship is not discussed. In particular, it is not clear to me why we would use a single coefficient when we are using a facotrized posterior approximation rather than a per-parameter one.

**Theoretical Claims:**

Not in detail.

---

> ### Author Rebuttal · Authors · 2025-04-01
>
> We appreciate your time for the review and respect your concern. However, we respectively argue our current experiments have sufficiently validated our contributions.
> ## Experiment, W2
> We believe our experimental design and results sufficiently support our claims for the following reasons:
> 1. We **have already compared various merging methods within the BECAME framework in Table 3**, as discussed in Line 366. Among these methods, CoFiMA is a per-parameter merging method based on Fisher Merging. The results show that our method **outperforms alternative merging strategies in both ACC and BWT**. This empirically supports the superiority of our merging coefficient and the adaptability of model merging in gradient projection (GP) methods. If there are any other baselines you would like us to compare, we are happy to include them.
> 2. The BECAME framework itself is a key contribution alongside the merging method, making its validation through experiments essential.
> 3. One of our primary motivations is to enhance the adaptability of existing GP methods, and the results in Tables 1 and 2 show that BACAME effectively integrates with various GP methods, significantly improving their performance.
> To further strengthen our study, we conduct experiments on applying model merging directly to CL without GP:
>
> ||10-S CIFAR100||20-S MiniImageNet||
> |---|---|---|---|---|
> ||ACC|BWT|ACC|BWT|
> |finetune|57.98|-20.19|57.06|-10.93|
> |1/t|60.69|-1.44|47.54|0.91|
> |CoMA|64.42|-9.79|60.99|-5.00|
> |CoFiMA|61.17|-0.38|62.26|-1.40|
> |Ours|**65.03**|-1.26|**64.75**|-2.11|
>
> These results further highlight the effectiveness of our merging coefficient compared to other methods.
> ## Claims&Theory&W3&Q2
> We respectfully argue that our theoretical analysis using Laplace approximation is valid for the following reasons:
> 1. Given the inherent complexity of deep learning optimization, which is influenced by numerous factors, it is impractical to analyze every possible case explicitly. Therefore, approximation techniques are **not only common but** **also** **essential for deriving meaningful theoretical** **insights****.** This is widely recognized in the field of machine learning.
> 2. The use of Laplace approximation in neural networks has been **well-established since 1992** (A Practical Bayesian Framework for Backpropagation Networks, cited by 4264), and numerous studies have built upon this foundational framework. Recent works (Lee et al., 2017; Marouf et al., 2024; Kirkpatrick et al., 2017; Ritter et al., 2018) have successfully **applied Laplace approximation in** **CL**, further validating its applicability
> 3. Model merging involves two key components: the model parameters {$\theta\_i$}$_n$ and their corresponding weights {$\lambda_i$}$_n$. In our setting we have {$\theta_i$}$\_n$={$\theta\_{t−1}^*,\theta_t$}, and our optimal merging coefficient is derived based on the merging trajectory between these two endpoints.
> 4. We have provided additional empirical evidence through loss function visualization (Figs. 1 and 2) and comparative experiments (Table 3), further supporting the reliability of our theoretical analysis.
> ## Relation&W1&Q1
> 1. Both Fisher Merging and our method are based on Laplace approximation and use Fisher for calculation, yet they differ significantly in many aspects. Below is a detailed comparison:
>
> ||Theoretical Basis|Usage of Fisher|Setting|Merging Coefficient $\lambda$|Number of Merges|Number of Models Merged|Relationship Between Models|Training Dataset|
> |---|---|---|---|---|---|---|---|---|
> |FisherMerging|Laplace Approx.|Weight importance of parameters|Ensemble, Finetune|Hyperparameter|1|n|Trained from a same initialization|Same|
> |Ours|Laplace Approx.|Calculate $\lambda$|CL|Adaptive|T(tasks count)-1|2|$\theta_{i+1}$ trained from $\theta_i$|Different for each task|
>
> Our method fills a crucial gap in existing merging approaches for CL by considering the influence of prior tasks on new task learning.
> 2. Using a scalar $\lambda$ rather than a vector is a deliberate choice for theoretical analysis. Ablation study and Table 3 also indicate that **per-parameter merging does not necessarily yield better results**. **Please refer to Q2** **of the** **response to reviewer H5hN for more details.**
>
>
> We hope these detailed responses and additional experiments address your concerns.

---

> > ### Comment · Reviewer_Nkj6 · 2025-04-08
> >
> > Thank you for the additional results and highlighting Tab 3. I hadn't taken the latter into account properly, apologies for the oversight. I will adjust my score.
> >
> > However, I never argued against Laplace or the use of approximations. My point is simply to use accurate language in describing methods and results. And the moment you use an approximation, you lose any guarantees of optimality or correctness. I agree that the approximations make sense and it's great that they work well empirically. However, that is not a justification for making loose/incorrect statements in a scientific paper.

---

> > > ### Author Response · Authors · 2025-04-09
> > >
> > > Thank you for your constructive feedback. We sincerely appreciate your recognition of our method and experimental results. It is encouraging for us to see our shared understanding of the value of applying Laplace approximation to analyze complex neural networks.
> > >
> > >
> > >
> > > We fully acknowledge that the main theoretical framework in our paper is built upon the Laplace approximation. We are truly grateful for your professional suggestions and your emphasis on scientific rigor. We also acknowledge that the language use is not accurate enough, which we will modify carefully.
> > >
> > > We have thoroughly reviewed expressions related to the **optimal merging coefficient in describing our method and results** throughout the paper. **We promise to revise** **all of them** **to explicitly reflect that the merging coefficient is derived based on the Laplace approximation and the optimality of the merging coefficient is also based on the** **approximation.** Specifically, we will improve phrases such as "optimal merging coefficient" with more precise wording like "optimal merging coefficient based on the Laplace approximation." For example:
> > >
> > > - In Line 24, we will improve "... derive a closed-form solution for the optimal merging coefficient" to "... derive a closed-form solution for the optimal merging coefficient **based on the Laplace approximation**."
> > > - In Line 88, we will refine the initial sentence as "**Based on the Laplace approximation,** we demonstrate that the ..."
> > > - In Line 59, "... derives a closed-form solution for the optimal coefficient **upon the Laplace approximation.**"
> > >
> > >
> > >
> > >
> > > Once again, thank you for your valuable feedback and for acknowledging the strengths of our approach and experiments. We hope that our proposed revisions address your concerns.

---

### Decision · Program_Chairs · 2025-05-01

**Decision:**

Accept (poster)

**Comment:**

This paper presents a framework designed to tackle the stability-plasticity trade-off in continual learning. This work uses a Bayesian approach to adaptive model merging, which aims to improve performance in continual learning by providing a closed-form solution for the merging coefficients. The paper's main strengths lie in its theoretical contributions, demonstrated through extensive empirical results on common benchmarks like CIFAR100 and MiniImageNet, showcasing improvements over existing methods. Additionally, the framework's compatibility with different gradient projection methods further amplifies its adaptability and potential for broader applicability.

The reviewers had varied scores initially: 2, 3, 4, 1. Much of the concern centred around the lack of clarity in theoretical claims and insufficient comparison with alternative merging methods. The authors provided clarifications and additional experiments during the rebuttal process, addressing these issues. Reviewer Nkj6 adjusted their score (1-2) after recognising that the initial evaluation overlooked the detailed discussion in Table 3, which contrasted various merging strategies. Despite a lot of prompting, there wasn't a whole lot of discussion, so it's hard to know if the initial concerns were fully covered.

The main weakness in the original submission was the lack of precision in theoretical claims about the optimality of the merging coefficients, which the authors acknowledged and promised to amend. There were also concerns regarding computational and memory overheads, but  the authors effectively demonstrated that their method remains competitive in terms of performance enhancements relative to this additional cost.

This approach is softer than trying to do fully Bayesian continual learning, which has been attempted many times and is problematic for various reasons. Here the Bayesian formulation is used only for merging models, and can be used with any gradient projection method. Considering the reasonable resolution of concerns and the contribution to Bayesian continual learning literature, I recommend accepting this paper to ICML.